# CMG2/ANTXR2 regulates extracellular collagen VI which accumulates in hyaline fibromatosis syndrome

Jérôme Bürgi[1], Béatrice Kunz[1], Laurence Abrami[1], Julie Deuquet[1], Alessandra Piersigilli[2,3], Sabine Scholl-Bürgi[4], Ekkehart Lausch[5], Sheila Unger[6], Andrea Superti-Furga[6], Paolo Bonaldo[7] & F. Gisou van der Goot[1]

Loss-of-function mutations in capillary morphogenesis gene 2 (CMG2/ANTXR2), a transmembrane surface protein, cause hyaline fibromatosis syndrome (HFS), a severe genetic disorder that is characterized by large subcutaneous nodules, gingival hypertrophy and severe painful joint contracture. Here we show that CMG2 is an important regulator of collagen VI homoeostasis. CMG2 loss of function promotes accumulation of collagen VI in patients, leading in particular to nodule formation. Similarly, collagen VI accumulates massively in uteri of $Antxr2^{-/-}$ mice, which do not display changes in collagen gene expression, and leads to progressive fibrosis and sterility. Crossing $Antxr2^{-/-}$ with $Col6a1^{-/-}$ mice leads to restoration of uterine structure and reversion of female infertility. We also demonstrate that CMG2 may act as a signalling receptor for collagen VI and mediates its intracellular degradation.

[1] Global Health Institute, Ecole Polytechnique Fédérale de Lausanne (EPFL), Lausanne 1015, Switzerland. [2] Comparative Mouse Physiology Platform, Faculty of Life Sciences, EPFL, Lausanne 1015, Switzerland. [3] Institute of Animal Pathology, Vetsuisse Faculty, University of Bern, Länggassstrasse 122, Bern 3012, Switzerland. [4] Medical University of Innsbruck, Clinic for Pediatrics I, Inherited Metabolic Disorders, Anichstrasse 35, 6020 Innsbruck, Austria. [5] Department of Pediatrics, University of Freiburg, Freiburg 79106, Germany. [6] Division of Molecular Pediatrics, Centre Hospitalier Universitaire Vaudois, University of Lausanne, Lausanne 1011, Switzerland. [7] Department of Molecular Medicine, University of Padova, Padova 35122, Italy. Correspondence and requests for materials should be addressed to F.G.v.d.G. (email: gisou.vandergoot@epfl.ch).

Hyaline fibromatosis syndrome (HFS) is a painful and disfiguring, potentially lethal, genetic disorder, the hallmark of which is the progressive growth of subcutaneous nodules, caused by accumulation of extracellular matrix (ECM). Patients also present gingival hypertrophy, severe and painful joint contracture, skin thickening and hyperpigmentation over the large joints[1,2]. HFS is caused by loss-of-function mutations in capillary morphogenesis gene 2 (refs 3,4) (CMG2), also known as anthrax toxin receptor 2 (ANTXR2). The *ANTXR2/CMG2* gene encodes a single-pass transmembrane protein harbouring an extracellular von Willebrand A (vWA) domain proposed to bind collagen IV and laminin[5]. The ANTXR2 name stems from its well-characterized role in binding and endocytosis of *Bacillus anthracis* toxins[6,7]. However, its physiological role is poorly understood. In zebrafish, it is involved in orienting cell division during embryogenesis within the planar cell polarity pathway[8]. In mice, *Antxr2* knockout was first reported to have no major consequence, other than resistance to anthrax infection[9]. Subsequent observations however showed that female *Antxr2*[−/−] mice suffer from severe progressive fibrosis of the uterus leading to disruption of tissue architecture and failure of parturition[10,11].

Here we discovered an uncharacterized function of CMG2 that can account for both the presence of nodules in HFS patients and the uterine phenotype in mice. We show that both the nodules of HFS patient and the uteri of *Antxr2*[−/−] female mouse are highly enriched in collagen VI. The progressive accumulation of collagen VI leads to loss of uterus integrity and ultimately sterility. Remarkably, we find that *Antxr2*[−/−]::*Col6a1*[−/−] double knockout mice are fertile, with complete restoration of the uterine myometrial layers. We go on to show that CMG2 may be a receptor for collagen VI and mediates its transport to lysosomes for degradation. Thus, loss of CMG2 function causes accumulation of collagen VI in the ECM, which with time may lead to tissue disruption and disease. The identification of CMG2 as collagen VI regulator opens new avenues towards understanding other processes involving this ECM protein such as obesity-induced insulin resistance[12,13] and muscle homoeostasis[14].

## Results

**Accumulation of collagen VI in HFS patient nodules**. Two biopsies of nodules, which often form at body site exposed to mechanical stress[15], were obtained from a 6-year-old HFS patient (Supplementary Fig. 1a and Supplementary Note 1 for patient information). One biopsy was taken from a nodule on the scalp on the posterior region of the head and the other from the back of the left ear, both with matching non-nodular adjacent tissue samples. Non-nodular tissues showed normal skin architecture, with a regular epithelial layer, and a connective tissue of the dermis composed of a thick fibrillar ECM with embedded vessels (Fig. 1a). Nodules contained very few, spindle-shaped cells embedded in an abundant amorphous, pale eosinophilic ECM (Fig. 1a) as reported for other patients[16,17]. Sirius red (SR) staining confirmed differences in ECM between normal and nodular tissues (Supplementary Fig. 1b). Finally, ultrastructure analysis by electron microscopy (EM) showed that whereas in the non-nodular tissue, cells were surrounded by a fibrillar ECM, the few cells observable in the nodule, were surrounded by a granular somewhat amorphous matrix with occasional fibrillar bundles (Fig. 1b).

SDS–PAGE analysis of protein extracts from these tissues revealed the presence of a single major 100 kDa band in nodular, but not in non-nodular, samples (Fig. 1c). Mass spectrometry (MS) analysis showed that it was composed of fragments of the three alpha chains of collagen VI (Fig. 1c)[18]. Further analysis showed that collagen α3 (VI) peptides were also present in higher and lower molecular weight bands of the gel, showing a,

respectively, wider and narrower peptide coverage, suggesting partial processing of collagen VI in the tissue or in the biopsy (Supplementary Fig. 2). Unbiased MS analysis of the total protein contents of the nodules showed that collagen VI accounted for more than 30% of the total peptides (Supplementary Fig. 1c) and that it was the only collagen for which the total peptide count increased between the nodular and the non-nodular tissues (Fig. 1d; data available on the PRIDE repository with the identifier PXD006268 and 10.6019/PXD006268). The MS analysis also indicated that while non-nodular tissues were enriched in cellular proteins, the nodules contained significant amounts of serum proteins (Supplementary Table 1), consistent with the blood vessels leakage proposed in HFS patients[19]. Drastic accumulation of collagen VI in nodules was confirmed by western blot analysis, which also showed moderate increase in collagen I and fibronectin (Fig. 1e).

To assess whether collagen VI accumulation was due to increased gene expression, we performed a qRT–PCR analysis on a panel of fibroblasts derived from different HFS patients (Supplementary Table 2). The mRNA levels for the α1, 2 and 3 chains of collagen VI (Fig. 1f) and α1 and 2 of collagen I (Supplementary Fig. 1d) were similar between control and HFS patient fibroblasts. Differences were observed in the mRNA levels for the α1 and 2 chains of collagen IV, but with no correlation to disease (Supplementary Fig. 1e). Thus, loss-of-function mutations in CMG2 do not lead to constitutive increase in collagen I, IV or VI gene expression.

**Collagen VI accumulation in the uteri of *Antxr2*[−/−] mice**. Fibrosis has also been reported for the uterus of *Antxr2*[−/−] mice[10,11]. To analyse the potential similarities between patient nodules and fibrotic mouse uteri, we generated *Antxr2*[−/−] mice (Supplementary Fig. 3a–c). In full agreement with the previous reports[10,11], our *Antxr2*[−/−] female mice displayed a progressive fibrosis of the uterus (observed in all $n = 19$ mice analysed). Haematoxylin–eosin (H&E) staining of uterus transverse sections showed a loss of tissue organization, with disruption of both myometrial layers and thickening of the endometrium, starting as early as 6 weeks after birth and becoming increasingly pronounced, as shown at 38 weeks (Fig. 2a). Staining with SR confirmed collagen accumulation in the disorganized uterine layers (Fig. 2a), reminiscent of HFS patient nodules. Ultrastructure EM analysis confirmed that the myometrium of *Antxr2*[−/−] mouse uteri was mostly composed of ECM, in comparison to that of *Antxr2*[+/+] mice, which was highly cellular (Fig. 2b, upper panels). In addition to being drastically more abundant, the ECM in the uteri of *Antxr2*[−/−] mouse was altered in its structure, containing microstructures reminiscent of collagen VI beaded filaments, while the ECM of wild-type (WT) mice uteri was composed of fibrillar structures (Fig. 2b, lower panel). We next performed immunogold labelling for collagen VI. Staining was specific and exclusively extracellular, in the uteri of both WT and *Antxr2*[−/−] mice (Fig. 2c and Supplementary Fig. 4). Interestingly, collagen VI labelling in WT mice uteri was observed on collagen fibres, while in *Antxr2*[−/−] mice uteri, collagen VI labelling was mostly confined to abundant non-fibrillar structures (Fig. 2c, lower panels). The abundance of collagen VI-containing ECM in *Antxr2*[−/−] mice uteri (Fig. 2c) suggests that the overall content of this collagen, at the tissue level, must be greatly increased when compared to the uteri of WT mice.

We therefore analysed protein extracts from mouse uteri by western blot using a polyclonal antibody (Supplementary Fig. 5a). Consistent with the EM analysis, a strong increase, with different relative variation between mice, was observed for all collagen VI

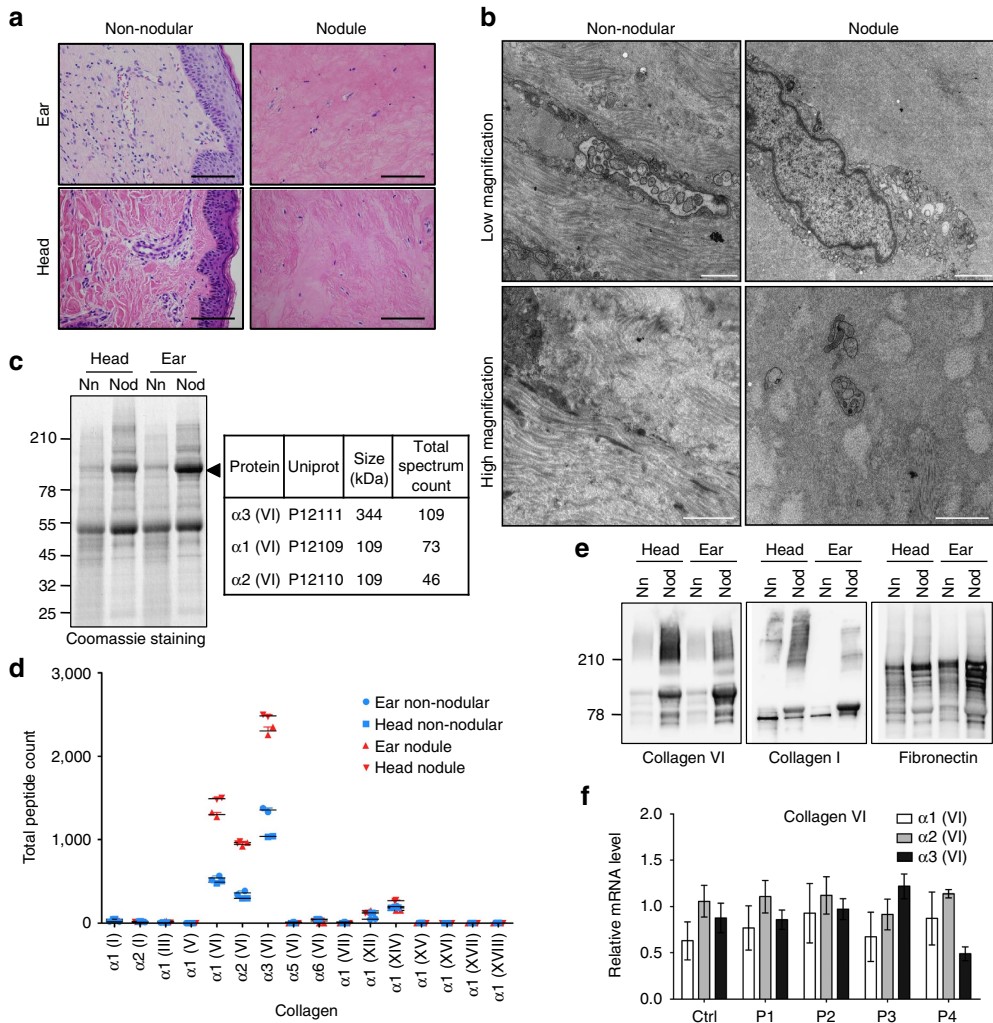

**Figure 1 | HFS patient nodules display ECM accumulation and loss of skin adnexa and dermal vessels.** (**a**) H&E staining of formalin-fixed non-nodular skin and nodule from the ear and the head of a HFS patient. The nodule displays a striking accumulation of ECM, causing secondary loss of skin adnexa, vascular structures and rarefaction of the dermal cellular component. One biopsy was obtained by tissue. A representative image is shown for each. Scale bar, 200 µm. (**b**) Ultrastructure by EM of non-nodular and nodular tissues of a HFS patient. Representative images of non-nodular or nodule tissues at low magnification (upper panel) and higher magnification (lower panel) are shown. At least two grid per tissue biopsies were prepared. Scale bar, 1 µm. (**c,e**) Comparison of non-nodular (Nn) and nodule (Nod) tissues from the ear and the head of a HFS patient. Tissue lysates were analysed by SDS–PAGE using 4–12% Bis-Tris gradient gels under reducing condition followed either by staining with Coomassie blue (**c**) or by western blot for collagen VI, collagen I or fibronectin (**e**). The prominent 100 kDa band (arrowhead) was excised, trypsin digested and its composition analysed by nano LC-MS/MS. Only the three collagen VI alpha chains were detected with >2 peptides. Migration of the molecular weight markers (in kDa) is indicated on the left. See Supplementary Fig. 7a for the uncropped version of the western blots. (**d**) Comparison of the complete tissue lysates from head and ear non-nodular and nodule analysed by nano LC-MS. The total number of peptides associated with each collagen protein is indicated. Each lysate was run twice in the nano-LC-MS to account for technical variability and both peptide counts are indicated. (**f**) Quantitative RT–PCR analysis of mRNAs coding for the α1, α2 and α3 chains of collagen VI in fibroblast cultures from unaffected control and four HFS patients (P1–P4) (error bars represent s.e.m.; $n = 3$).

chains, including the alternative α4, α5 and α6 (VI) (Fig. 3a). In contrast, there was no significant change in collagens I and IV (Supplementary Fig. 5b).

Since fibrosis often involves the transforming growth factor beta (TGFβ) pathway[20,21], we probed for its activation by measuring the mRNA levels of genes regulated by this pathway. We observed a 1.7-fold increase in TGFβ1 mRNA, as well as a twofold change of in the tumour necrosis factor transcript in the uteri of $Antxr2^{-/-}$ mice compared to control mice (Fig. 3b). There was, however, no concomitant increase in transcript levels of TGFβ-target genes such as the metalloproteases MMP14, 2, 8 and 9, alpha smooth muscle actin, Col3a1 or lysyl oxidase (LOX) (Fig. 3b). Thus, the accumulation of collagen VI does not appear to be a consequence of the activation of the TGFβ pathway.

We next analysed the mRNA levels of various collagen genes. qRT–PCR analysis of WT and $Antxr2^{-/-}$ mouse uteri showed that the mRNA expression of none of the alpha chains of collagen VI changed in response to $Antxr2$ knockout (Fig. 3c). The transcripts for the α1 and α2 chains of collagens I and IV also remained unchanged (Fig. 3d). Thus, the accumulation of collagen VI in the uteri of $Antxr2^{-/-}$ mice cannot be explained by transcriptional changes of ECM components or in the TGFβ pathway.

**CMG2 and the activity of extracellular metalloproteinases.** The accumulation of collagen VI in patient nodules and mouse uteri on loss of CMG2 function, without concomitant increases in

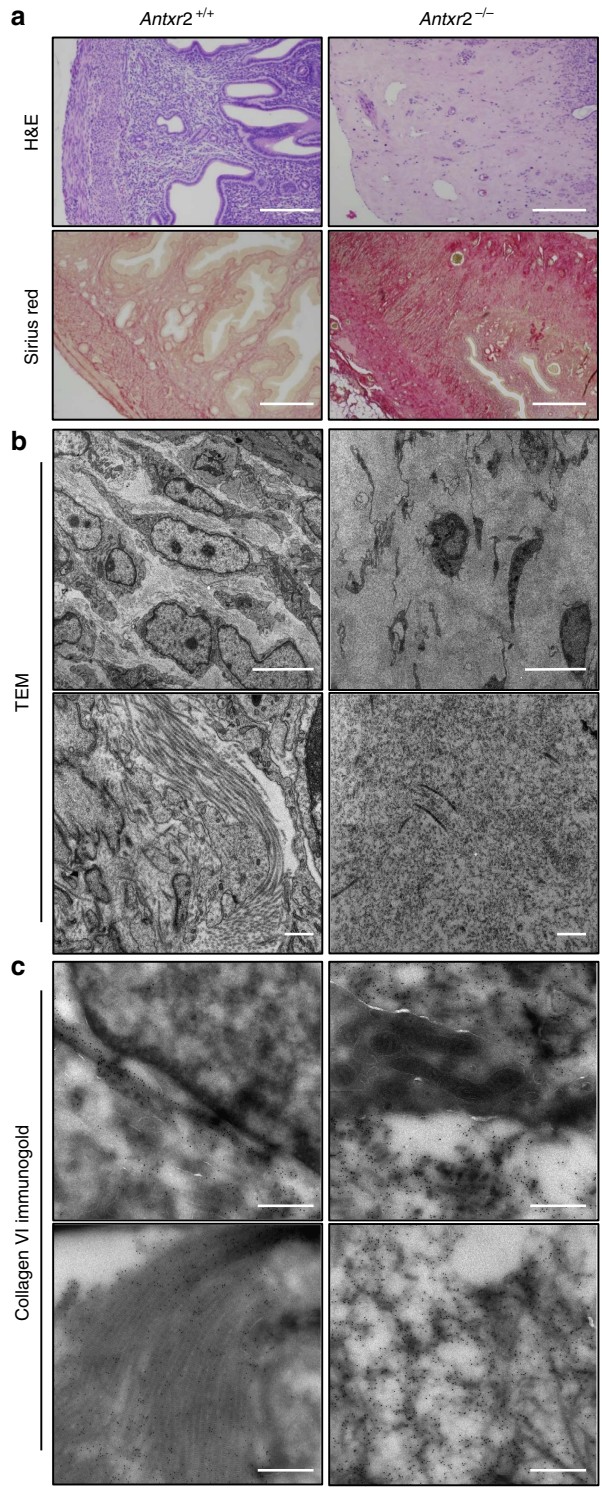

**Figure 2 | Structural changes in the uteri of *Antxr2*[−/−] mice. (a)** H&E and SR staining of formalin-fixed uterine tissues from 38-week-old WT (*Antxr2*[+/+]) and *Antxr2* knockout (*Antxr2*[−/−]) mice. *Antxr2* knockout mice display ECM accumulation with disarray, disruption and loss of the longitudinal and circular myometrial layers of the uterus. Representative image of at least *n* = 7 mice per genotype. Scale bar, 200 μm. **(b)** Ultrastructure of *Antxr2*[+/+] and *Antxr2*[−/−] uterine myometrial layer analysed by transmission electron microscopy (TEM). *N* = 2 mice. Scale bar, 5 μm (upper panels) and 500 nm (lower panels). **(c)** Collagen VI immunogold labelling on the *Antxr2*[+/+] and *Antxr2*[−/−] uterine myometrial layer analysed by TEM. *N* = 2 mice. Scale bar, 500 nm.

mRNA, indicates that its degradation, rather than its synthesis, must be defective. We therefore monitored the activity of extracellular metalloproteases (MMP) in patient nodules and fibroblasts, in mouse uteri and in tissue cultured cells. In the nodule biopsies, MMP14, a membrane-bound MMP known to activate the collagenase MMP2 (ref. 22), was mainly found in its inactive pro-form. Consistently, MMP2 was also inactive (Fig. 4a). This inactivation does not appear to be a direct consequence of CMG2 loss of function since MMP2 activity, analysed by gelatin zymography, was similar for fibroblasts derived from HFS and control patients (Fig. 4b).

We next performed a similar analysis on mouse uteri. In contrast to what was observed for the patient nodules, the activities of both MMP14 and MMP2 were higher in the uterus of *Antxr2*[−/−] mice than in that of control mice (Fig. 4c,d). To address the issue in a more controlled system, we analysed the effect of CMG2 silencing on MMP2 activity in tissue culture cells. CMG2 knockdown in RPE1 cells led to a significant increase in the MMP14 protein level accompanied by a twofold increase in MMP2 gelatinase activity (Fig. 4e,f).

Altogether, these observations indicate that there is no direct correlation between the function of CMG2 and the MMP14–MMP2 activities in affected tissues and thus that collagen VI accumulation is unlikely to be caused by reduced degradation by extracellular metalloproteases. Consistently, while the α3 (VI) chain alone, which is retained intracellular, is sensitive to degradation by MMP2 *in vitro*[23], full collagen VI fibrils were reported to be insensitive to MMPs degradation[24].

**Collagen VI drives uterine fibrosis in *Antxr2*[−/−] mice.** To again further insight into the functional link between CMG2/ANTXR2 and collagen VI, we generated *Antxr2*[−/−]::*Col6a1*[−/−] double knockout mice. In agreement with previous reports, our *Antxr2*[−/−] female mice were infertile, due to a parturition defect[10]. This can be explained by the perturbation of the uterus architecture (Fig. 2) and loss of contractile myometrial elements, which are known to cause drastic changes in the mechanical properties of the uterus[25]. In contrast, *Col6a1*[−/−] females are fully fertile[26]. The double knockout mice were viable with no obvious morphological defects. Quite remarkably, *Antxr2*[−/−]::*Col6a1*[−/−] females were fully fertile, with 100% (6/6) of the mice being able to give birth, as opposed to 0% (described in two studies[10,11] and confirmed here for two females) for *Antxr2*[−/−]::*Col6a1*[+/+] mice. *Antxr2*[−/−]::*Col6a1*[−/−] females were even able to undergo multiple pregnancies with normal litter size. Consistently, H&E and SR staining showed that the uteri of 15-week-old *Antxr2*[−/−]::*Col6a1*[−/−] mice in metestrus had a normal structure of muscle and endometrial layers, with no detectable accumulation of ECM (Fig. 5 and Supplementary Fig. 5c). The absence of collagen VI (Supplementary Fig. 5d) was not accompanied by increase in collagen IV and fibronectin (Supplementary Fig. 5e). Thus, genetic ablation of collagen VI in *Antxr2*[−/−] mice was sufficient to rescue the uterine phenotype. This striking recovery of the fertility indicates that the defect in uterus function is secondary to the accumulation of collagen VI, which caused disruption of the myometrial layers.

**CMG2 mediates collagen VI uptake and targeting to lysosomes.** The ability of the *Col6a1* knockout to restore tissue function in the *Antxr2*[−/−] mouse uterus points to a direct connection between CMG2 and collagen VI. Since CMG2 loss of function does not lead to an increase in collagen VI gene expression either in HFS patient fibroblasts or *Antxr2*[−/−] mice uteri, we investigated whether CMG2 can physically interact with this ECM protein. We first tested *in vitro* binding of the CMG2 vWA

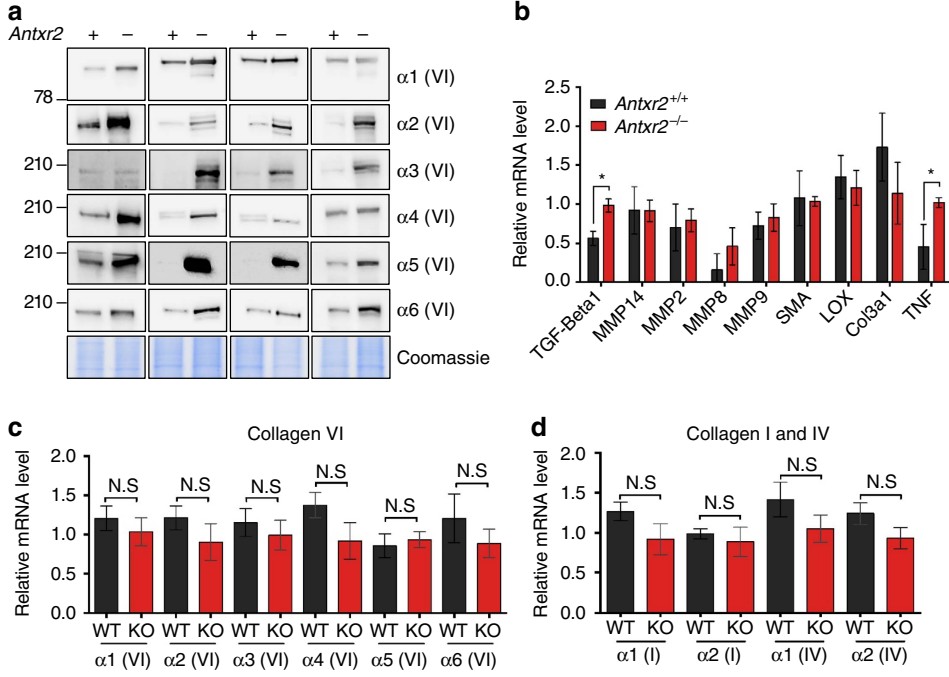

**Figure 3 | Severe collagen VI accumulation in the uteri of *Antxr2*<sup>−/−</sup> mice.** (**a**) Uterus lysates (20 μg) from littermate WT and *Antxr2* knockout (KO) mice were analysed by SDS–PAGE using 4–12% Bis-Tris gradient gels under reducing condition and western blotting against all the collagen VI alpha chains. The signals obtained varied beyond the dynamic range allowed with western blotting and thus the amplitude of the increase was not evaluated for the various alpha chains. Coomassie staining was used as loading controls. Migration of the molecular weight markers (in kDa) is indicated on the left. $N = 4$ mice. See Supplementary Fig. 7b for the uncropped version of the western blots. (**b**) Quantitative RT–PCR analysis of genes involved in the TGFβ pathway was performed on uteri from WT (*Antxr2*<sup>+/+</sup>) and *Antxr2* knockout (*Antxr2*<sup>−/−</sup>) mice. (Error bars represent s.e.m.; $n = 3$; *$P < 0.05$; two-tailed unpaired *t*-test). (**c,d**) Quantitative RT–PCR analysis of mRNAs coding for the α1 and α2 chains of collagens I, IV and VI in WT (WT) and *Antxr2* KO (KO) mouse uterus. (Error bars represent s.e.m.; $n = 6–7$; N.S., $P > 0.05$; two-tailed unpaired *t*-test).

domain with collagen VI. Missense mutations in the metal ion-dependent-binding domain (MIDAS) of the CMG2 vWA domain have indeed been found to trigger the most severe form of HFS, with death in early childhood[1,4]. We produced recombinant glutathione *S*-transferase (GST)-tagged CMG2 vWA domains, either WT or carrying the MIDAS disrupting D50A mutation[27] and probed their binding to a panel of commercially available purified ECM proteins by ELISA. At a difference from a previous study reporting that the CMG2 vWA domain binds collagen IV and laminin[5], we found that the CMG2 vWA domain preferentially binds collagen VI, in a MIDAS-dependent manner (Fig. 6a). The discrepancy between the two findings might be due to the difference in readout of the ELISA assays and the fact that ECM proteins were coated onto the ELISA plates in our approach, as opposed to the vWA domain in the previous study.

We next investigated whether full-length CMG2 was able to bind collagen VI in a cellular context and whether this elicits signal transduction. We have previously shown that when the anthrax toxin protective antigen (PA) binds to CMG2, src-like kinases are activated and phosphorylate CMG2, triggering its endocytosis[6]. To test whether collagen VI could elicit a similar response, HeLa cells transiently expressing WT or D50A CMG2 were plated onto dishes coated with different ECM proteins and CMG2 tyrosine phosphorylation was monitored. Phosphorylation of WT CMG2, but not of D50A, was observed on plating cells on collagen VI or on anthrax PA, whereas no significant CMG2 phosphorylation occurred when cells were plated on collagen I, collagen IV, laminin or fibronectin (Fig. 6b). We next tested whether subsequent events involved in PA internalization were also triggered on collagen VI binding. β-arrestin is an adaptor protein required for the anthrax

toxin-induced ubiquitination and endocytosis of CMG2 (ref. 28). As previously, cells were plated onto dishes coated with specific ECM proteins, β-arrestin was immunoprecipitated and we probed for the pulldown of CMG2. An interaction was only observed for cells plated on PA or collagen VI-coated dishes (Fig. 6c). Thus, CMG2 is able to bind collagen VI, and more specifically its triple helical domain, which is the domain present in the commercially available purified collagen VI obtained by pepsinization of human tissues. This binding event leads to intracellular signals such as the src-dependent phosphorylation of CMG2 cytoplasmic tail and the recruitment of β-arrestin.

The extracellular accumulation of collagen VI in human and mouse tissues on CMG2 loss of function, combined with the knowledge that anthrax toxin-induced CMG2 src phosphorylation and β-arrestin recruitment lead to endocytosis and transport to the late endocytic pathway, led us to test whether CMG2 can mediate collagen VI endocytosis and degradation. Purified collagen VI tetramers were added to RPE1 cells in which CMG2 expression was silenced or not by stable expression of an anti-CMG2 shRNA vector. We observed that the collagen VI complex associated with control cells and underwent degradation as a function of time, only 20% remaining after 6 h (Fig. 6d). This degradation could be inhibited by treating cells with the vATPase inhibitor bafilomycin A, indicating that collagen VI undergoes degradation in lysosomes (Fig. 6d and Supplementary Fig. 6a). Strikingly, while the collagen VI complex associated with shRNA CMG2 knockdown RPE1 cells, there was no sign of degradation even after 6 h (Fig. 6d). Thus, CMG2 is required for the intracellular degradation of collagen VI.

This was further confirmed using primary human fibroblasts from patient P3, who carried a frameshift mutation in exon 9, leading to a premature stop codon and non-sense-mediated

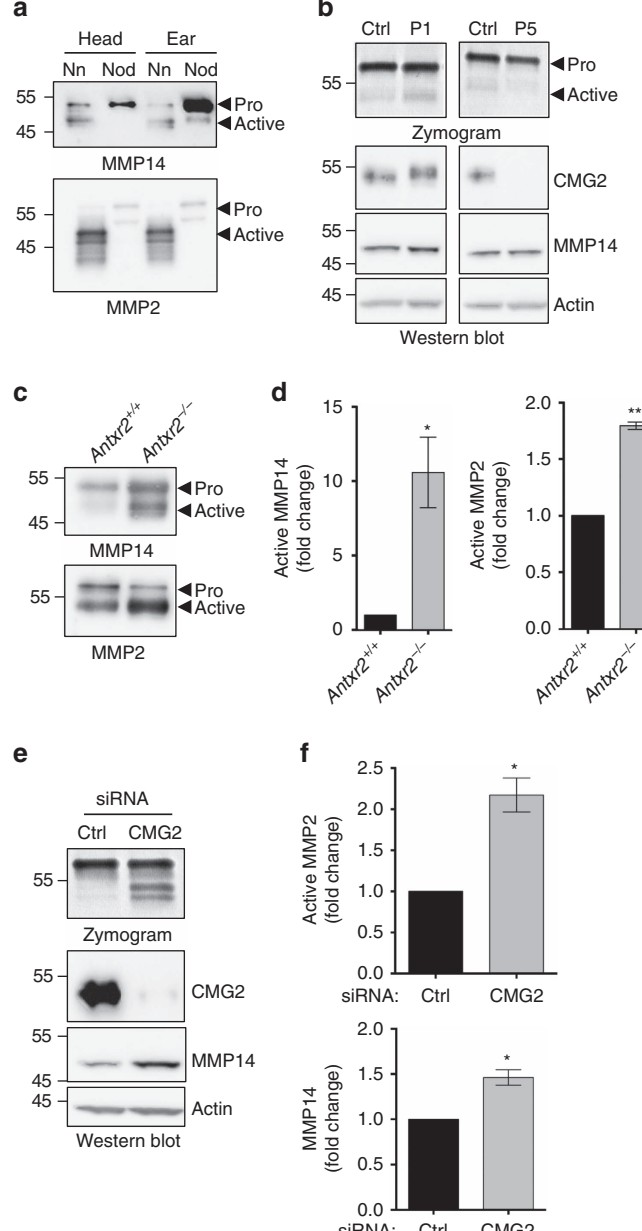

**Figure 4 | Regulation of MMPs activity on CMG2 loss of function.** (**a**) Lysates from non-nodular (Nn) and nodule (Nod) tissues from the ear and the head were analysed by SDS–PAGE and western blotting against MMP2 and MMP14. (**b**) Supernatant of HFS patients or control fibroblasts were normalized to protein concentration and analysed by gelatin zymography. Cell lysates were analysed by SDS–PAGE and western blotting against MT1-MMP, CMG2 and actin as loading control. (**c**) Uterus lysates from $Antxr2^{+/+}$ or $Antxr2^{-/-}$ mice were analysed by SDS–PAGE and western blotting against MMP2 and MMP14. (**d**) Quantification of active MMP2 and MMP14 by densitometric analysis. (**e**) RPE1 cells were treated with control siRNA or siRNA against CMG2 for 72 h. Cell supernatant were then normalized to protein concentration and used for enzyme zymography. Cell lysates were analysed by SDS–PAGE and western blotting against CMG2, MMP14 and actin as a loading control. (**f**) Quantification of active MMP2 and MMP14 by densitometric analysis. (**d**,**f**) Error bars represent s.e.m.; $n=3$; $*P<0.05$; $**P<0.01$; two-tailed unpaired $t$-test compared to control. See Supplementary Fig. 8a–d for the uncropped version of the western blots.

mRNA degradation (Supplementary Fig. 6b). As observed for RPE1 cells, the collagen VI complex associated with control patient cells and underwent degradation as a function of time, with only 35% of the initial level detectable after 6 h (Fig. 6e). Strikingly, P3 fibroblasts were unable to degrade collagen VI, even if the complex was found associated with cells. Thus, HFS patient fibroblasts essentially devoid of CMG2 protein are unable to degrade collagen VI. Loss of CMG2 does not have a pleiotropic effect on the endocytic pathway since internalization and lysosomal degradation of mouse IgGs was unaffected by CMG2 knockdown (Supplementary Fig. 6c).

Altogether, these observations show that CMG2 is a bona fide collagen VI receptor, which has the capacity to target its ligand for degradation. So far, no specific receptor for collagen VI has been identified. Collagen VI can bind several integrins, such α1, α2, α5, α10 and α11, as well as to the cell surface proteoglycan CSPG4/NG2 (ref. 29), but not as an exclusive ligand. As these proteins, CMG2 binds to the triple helical domain of collagen VI (Fig. 6a). This differs from its related protein ANTXR1/TEM8, which was reported to bind the C5 domain of collagen VI, a domain that is cleaved during collagen VI maturation[30].

## Discussion

Analyses of HFS patient nodules and of $Antxr2^{-/-}$ mouse uteri pointed towards a non-genetic interaction of CMG2 with collagen VI. Studies on purified proteins and tissue culture cells revealed that collagen VI can bind to the vWA domain of CMG2 via its triple helical domain. While a complete characterization of CMG2 interaction with collagen VI, including the specific-binding site and affinity, requires further studies, we found that this interaction leads to ligand-induced endocytosis of the receptor, which targets collagen VI to lysosomal degradation. These findings are consistent with previous observations showing that collagen VI fibrils are insensitive to MMPs[24] but require serine and cysteine hydrolases, such as those found in lysosomes, for degradation[31].

These findings provide an explanation for the collagen VI accumulation observed in certain tissues or body parts of HFS patients, leading to progressive loss of tissue integrity and thereby function. Why certain tissues are particularly sensitive to loss of CMG2 function requires further investigation. The sensitivity of the mouse uterus might be due to the fluctuating level of collagen VI in this organ, in particular during early gestation[32].

While excess of collagen VI is detrimental, decreased levels also lead to diseases, such as Ullrich congenital muscular dystrophy and Bethlem myopathy (OMIM #25090, #158810)[33,34]. Thus, the amount of collagen VI in the extracellular environment needs to be tightly controlled, and CMG2 appears to play a critical role in this process.

Binding of collagen VI to CMG2 might not systematically lead to endocytosis and degradation. An equilibrium between binding/signalling and endocytosis must exist, which depends on the concentration of collagen VI in the cellular environment, as described for the EGF receptor[35]. Also, the massive accumulation of collagen VI in the extracellular environment might not be entirely due to the absence of endocytosis. Since collagen VI was found to modulate MMPs activity[10,36], its accumulation could sequester inactive MMPs creating a positive feedback loop of ECM accumulation, leading to fibrosis[37].

The data obtained in $Antxr2^{-/-}::Col6a1^{-/-}$ mice indicates that collagen VI is the key initiator of the uterus fibrosis and most likely of nodule formation in humans. While inhibition of collagen synthesis has so far not been evaluated as a strategy to treat fibrotic disease[38], specific targeting of collagen VI could be a new therapeutic strategy for HFS patients to attenuate or even

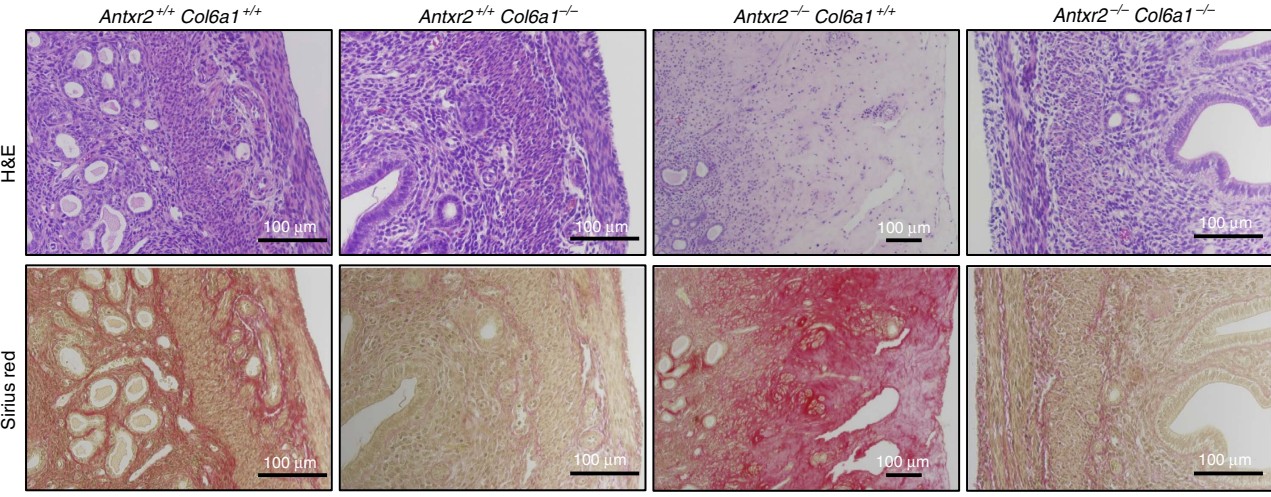

**Figure 5 | Deletion of Col6a1 is sufficient to rescue the uterine phenotype of Antxr2$^{-/-}$ mice.** H&E and SR staining of uterine tissues from 15-week-old Antxr2$^{+/+}$::Col6a1$^{+/+}$, Antxr2$^{+/+}$::Col6a1$^{-/-}$, Antxr2$^{-/-}$::Col6a1$^{+/+}$ and Antxr2$^{-/-}$::Col6a1$^{-/-}$ mice in metestrus. Failure of parturition in Antxr2$^{-/-}$::Col6a1$^{+/+}$ mice is caused by the same mechanisms described for Antxr2$^{-/-}$ (ref. 10), and this phenotype is reversed after ablation of collagen VI in Antxr2$^{-/-}$::Col6a1$^{-/-}$ mice. Representative image of at least $n = 3$ mice per genotype. Scale bar, 100 μm.

revert the formation of nodules. As a novel collagen VI regulator, it will also be of great interest to investigate the role of CMG2 in the collagen VI-dependent regulatory processes such as muscle homoeostasis[14] and adipogenesis[12].

## Methods

**Human samples.** Human tissues, for analyses or derivation of fibroblasts, were obtained after patient surgery, with explicit written consent of the patients and/or their legal tutors. Fibroblasts derived from HFS patients were included in a biobank dedicated to HFS and affiliated to the Lausanne University Hospital biobank. Our study followed the tenets of the declaration on Helsinki and was approved by the ethical committee of the Lausanne University Hospital (authorization A070055).

**Generation of Antxr2$^{-/-}$ and Antxr2$^{-/-}$::Col6a1$^{-/-}$ mice.** CMG2 knockout mice were generated by the Mouse Clinical Institute (Strasbourg, France) following a routine protocol. Mouse ES cells (MSI-129Sv/Pas) were electroporated with a 13 kb targeting vector with 5′ and 3′ homology arms targeting Antxr2 exon 3. A neo$^R$ cassette was used for G418 selection of positive clones, and a first PCR screen on the loxP site and the 3′ homology arm was performed. Homologous recombination was determined by Southern blotting with neo$^R$ and Antxr2 3′ probes. ES cell clones were then injected into blastocysts and implanted into pseudopregnant mice. Breeding of chimeric mice to achieve germ-line transmission was performed. The neo$^R$ cassette was removed by flippase-mediated deletion (FLP). Heterozygous (Antxr2$^{+/-}$) mice were generated by Cre-mediated recombination following breeding with Nestin-Cre mice. The mice were backcrossed for 10 generations onto the C57Bl6/J genetic background (Charles River Laboratories). Antxr2$^{+/+}$ mice were kept in a specific pathogen free (SPF) environment with 12 h light and 12 h dark cycle.

Col6a1$^{-/-}$ mice were generated by targeted deletion of the gene exon 2 (ref. 26). Col6a1 mutant mice were generated in the C56Bl6/N background and backcrossed with WT animal at least 10 times. Antxr2::Col6a1 double-mutant mice were generated by crossing heterozygous animals. Antxr2::Col6a1 were housed in a conventional facility on a 12 h light/dark cycle. Antxr2::Col6a1 double-mutant mice were always used at the F1 generation and were a mix of the C56Bl7/J and N substrains. Experiments were all performed using littermate from the different genotypes.

Depending on the experiment, female mice were used from 15 weeks up to 38 weeks.

For animal experimentation, all procedures were performed according to protocols approved by the Veterinary Authorities of the Canton Vaud and according to the Swiss Law (licence VD 2144.4, EPFL).

**Genotyping.** Mice were genotyped by PCR amplification using genomic DNA extracted from the tail. Several set of primers depending on the gene to test were used: for the Antxr2 knockout alleles, forward primer 5′-GGTGACTTTTGTCT GGACTCTTATC-3′ and reverse primer 5′-CAGATACGTGGATGGTTGC-3′. For the genotyping of the Col6a1 knockout alleles, the forward primer 5′-CTGCTG GTGAGAATGGATGGTGT-3′ and reverse primer 5′-TGTGCCGAGTCATAG CCGAATAG-3′ were used.

**Histology.** Mouse tissues were fixed in 4% paraformaldehyde overnight at 4 °C and embedded in paraffin. Serial sections (4 μm) were stained with H&E and SR. The slides were then analysed in a blinded fashion by a board-certified veterinary pathologist.

Mouse uteri were analysed during the metestrous phase. Indeed, uterine tissues change during the menstruous cycle and can potentially mimic cellularity rarefaction and ECM expansion, such as oedema. Such changes are however minimal during the metestrous phase.

**EM.** Mice were perfused via the heart with a buffered mix of 2.5% glutaraldehyde and 2% paraformaldehyde in phosphate buffer (0.1 M, pH 7.4). The animal was left for 2 h and then 100 μm thick sections cut in the transverse plane through the middle portion of the uterus using a vibratome. These slices were then postfixed with 1.5% potassium ferrocyanide and 1% osmium tetroxide, then with 1% osmium tetroxide alone, and finally in 1% aqueous uranyl acetate. They were dehydrated with increasing concentrations of alcohol, embedded in Durcupan resin and hardened at 65° for 24 h. A segment of the uterine wall was then thin sectioned using a diamond knife at a thickness of 50 nm and mounted onto single slot copper grids with a pioloform support film, stained with lead citrate and uranyl acetate, and imaged at 80 kV in a TEM (Tecnai Spirit, FEI Company).

**Cryo-immuno EM.** To localize collagen VI within the different compartments of the uterine wall, we used cryo-immuno EM. Mice were perfused via the heart with a buffered solution of 4% paraformaldehyde and the uterus was then removed and sliced transversally with a razor blade. These sections were then infiltrated with a 2.3 M solution of sucrose, and frozen in liquid nitrogen onto aluminium mounting pins and stored. Cryo sections were cut at − 90 °C with a diamond knife at 90 nm thickness and collected onto formvar support films on copper grids, these were then brought to room temperature ready for immuno-staining. The grids were incubated overnight with primary antibody, washed and then exposed for 3 h to a secondary antibody carrying 10 nm gold particles. After washing, and fixation with buffered glutaraldehyde, the grids were next stained with uranyl acetate, followed by uranyl acetate in methylcellulose on ice, and then air dried ready for imaging with transmission EM.

**Antibodies and reagents.** Rabbit anti-collagen I antibody was purchased from Rockland antibodies (#600-401-103-0.5) and used at 1:2,000. Rabbit anti-collagen IV (ab6586) and collagen VI (ab6588) antibodies were from Abcam and used at 1:3,000 dilution. The antibodies against the alpha chains of mouse collagen VI were a kind gift of Raimund Wagener[39] and used at 1:500. Rabbit anti-fibronectin antibody was from Sigma (F3648) and used at 1:1,000 dilution. Rabbit anti-MMP2 antibody was from Abcam (ab37150) and used at 1:2,000 dilution. Mouse anti-MMP14 antibody was from Millipore (MAB3328) and used at 1:3,000 dilution. Rabbit β-arrestin 1 antibody was from Thermo Fischer Scientific (PA5-19582) and used at a 1:1,000 dilution. Mouse anti-actin antibody was

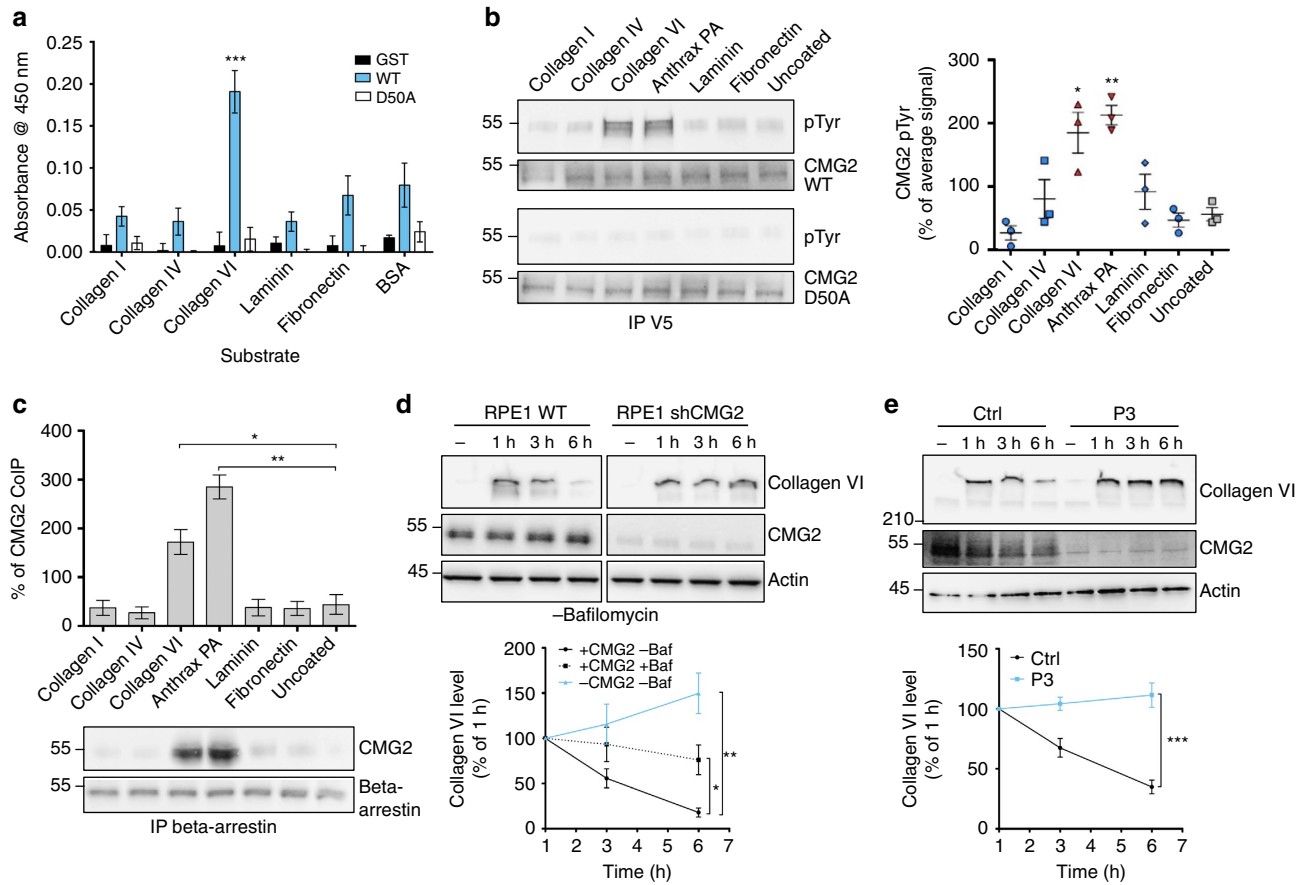

**Figure 6 | Collagen VI binding to CMG2 leads to signalling and degradation.** (**a**) Collagen I, collagen IV and collagen VI, fibronectin, laminin, PA and BSA were coated on 96-well plates. Subsequently, purified GST, WT CMG2 vWA-GST (WT) or D50A mutant CMG2 vWA-GST (D50A) were incubated at $1 \mu g \mu l^{-1}$ for 2 h at 25 °C and the interaction revealed using an antibody against GST and a horseradish peroxidase-coupled secondary antibody. The reaction was quantified with a plate reader at 450 nm. (**b**) HeLa cells were transfected for 48 h with a C terminally V5-tagged WT or D50A mutant CMG2 (D50A), and subsequently plated onto different dishes coated or not with either collagen I, collagen IV, collagen VI, PA, laminin or fibronectin. Cells were collected 60 min after plating. V5 immunoprecipitates were then analysed by SDS–PAGE using 4–12% Bis-Tris gradient gels under reducing condition and western blotting for phospho-Tyr and V5. The phospho-Tyr signal after immunoprecipitation against the V5-tag was quantified by densitometric analysis, and the values normalized as a percentage of the average signal. (**c**) RPE1 cells were plated on dishes coated with collagen I, collagen IV and collagen VI, PA, laminin, fibronectin or on uncoated dishes. Cells were collected 60 min after plating. β-arrestin immunoprecipitates were then analysed by SDS–PAGE using 4–12% Bis-Tris gradient gels under non-reducing condition and western blotting for endogenous CMG2 and β-arrestin. The CMG2 signal was quantified by densitometric analysis. (**d–e**) CMG2 knockdown RPE1 cells (**d**) or control and P3 fibroblasts (**e**) were blocked with 5% bovine serum albumin 30 min before addition of purified collagen VI tetramer at $1 \mu g \, ml^{-1}$. Cells were collected 1, 3 or 6 h later. Collagen VI degradation was assessed by SDS–PAGE using 4–12% Bis-Tris gradient gels under non-reducing condition and western blotting for collagen VI, endogenous CMG2 and actin as a loading control. (**a–e**) Error bars represent s.e.m.; $n = 3$; *$P < 0.05$; **$P < 0.01$; ***$P < 0.001$; two-tailed unpaired $t$-test to BSA (**a**), uncoated (**b,c**) or control (**d,e**). See Supplementary Fig. 8e–h for the uncropped version of the western blots.

from Millipore (MAB1501) and used at a 1:3,000 dilution. Anti-phospho-Tyr (clone 4G10) was from Upstate and used at 1:500. Mouse anti-V5 antibody were purchased from Invitrogen and used at a 1:2,000 dilution and rabbit anti-V5 antibody from Covance and used at a 1:3,000 dilution. Anti-mouse or human CMG2/ANTXR2 are monoclonal antibodies developed by rat genetic immunization and purified from hybridomas supernatant (Genovac) both used at 1:2 dilution. Horseradish peroxidase-conjugated secondary antibodies were from Pierce and used at 1:2,000 dilution. Protein G beads were purchased from GE Healthcare.

**Patient tissue preparation and MS analysis.** HFS patient tissue samples of similar weight were cut with a scalpel in 200 μl of lysis buffer (2% SDS in Tris-HCl 100 mM). After processing into very small pieces, another 200 μl of lysis buffer were added. The suspended tissues were then put in a Lysing Matrix tube (MP Biomedicals) and homogenized in a tissue lyser (Qiagen) 3 × for 5 min, with 2 min resting period at 4 °C between each round. Homogenates were then pelleted at 5,000 r.p.m. for 3 min at 4 °C. Protein amount was quantified by BCA assay (Pierce). After protein concentration normalization, DTT was added to achieve a final concentration of 50 mM. Protein concentrations of homogenates were determined by BCA (Pierce) and equal amount of proteins were analysed by SDS–PAGE using 4–12% Bis-Tris gradient gels under reducing condition followed

by Coomassie staining or western blotting, or 20 μg in 20 μl (final concentration $1 \mu g \mu l^{-1}$) of the homogenates were analysed by nano LC-MS/MS. Each sample was digested by filter-aided sample preparation[40]. Peptides were desalted on C18 StageTips[41] and dried down by vacuum centrifugation before mass spectromic analysis. For LC-MS/MS analysis, peptides were resuspended and separated by reversed-phase chromatography on a Dionex Ultimate 3000 RSLC nanoUPLC system in-line connected with an Orbitrap Q Exactive Mass-Spectrometer (Thermo Fischer Scientific). Each homogenate was analysed twice ($2 \times 10 \mu l$) to assess technical reproducibility. For protein identification from Coomassie staining, the relevant protein band was excised and trypsin digested before identification with nano LC-MS/MS.

**MS data analysis.** Database search was performed using Mascot, MS-Amanda and SEQUEST in Proteome Discoverer v.1.4 against a UniProt human database release 2014_06. All searches were performed with trypsin cleavage specificity, up to two missed cleavages allowed and ion mass tolerance of 10 p.p.m. for the precursor and 0.05 Da for the fragments. Carbamidomethylation was set as a fixed modification, whereas oxidation (M), acetylation (Protein N-term), phosphorylation (STY) were considered as variable modifications. Data were further processed and inspected in the Scaffold 4 software (Proteome Software).

**ELISA assay.** WT and D50A CMG2 vWA domains were cloned in pGex-4T-1 plasmid (GE Healthcare) and expressed in *E. coli* as GST fusion proteins. For ELISA, 1 µg of collagen I (Corning ref. 354243), collagen IV (Corning ref. 354245), collagen VI (Corning 354261), laminin (Sigma L6274), fibronectin (Corning ref. 354008) and BSA in 100 µl PBS were coated for 2 h at room temperature on Maxisorp 96-well plates (NUNC). Samples were treated for 30 min with TBST (TBS pH 7.5 with 0.1% Tween) and then incubated for 2 h at room temperature with 0.1 µg of recombinant GST-vWA protein in 100 µl of TBST–BSA (TBS pH 7.5 with 0.1% Tween, 1% BSA, 1 mM CaCl$_2$, 1 mM MgCl$_2$). Samples were then washed with TBST–BSA and incubated for 1 h with an anti-GST antibody in TBST–BSA. After a TBST–BSA wash, samples were incubated for 1 h with a secondary antibody coupled to horseradish peroxidase in TBST–BSA. Levels of bound GST-vWA were measured by adding 0.4 mg ml$^{-1}$ o-phenylene-diamine (Sigma) to the samples and reading the peroxidase activity at 405 nm on a 96-well plate reader (SpectraMax M2, Molecular Devices).

**Cells and plasmids.** HeLa cells (ATCC) were grown in modified eagle's medium (Sigma Life Science) supplemented with 10% fetal calf serum, 2 mM L-glutamine, non-essential amino acids, penicillin and streptomycin (GIBCO).

Primary fibroblasts were isolated after incubating patient biopsies 5 min at 37 °C in trypsin, followed by trypsin inactivation in Dulbecco's modified eagle medium (Sigma Life Science) supplemented with 10% fetal calf serum, penicillin and streptomycin (GIBCO). Dissociating fibroblasts where then amplified and subsequently frozen in complete medium with 10% dimethylsulfoxide and kept in liquid nitrogen. Fibroblasts where used until passage 6. All cell lines were tested for mycoplasma contamination.

Human CMG2 (isoform 4, Uniprot identifier P58335-4) was cloned in a pcDNA3.5/V5-HIS-TOPO expression vector.

**siRNA and cDNA transfections and phosphorylation assay.** CMG2 siRNA was purchased from Qiagen. As a control siRNA, the following target sequence of the viral glycoprotein VSV-G: 5′-attgaacaaacgaaacaagga-3′ was used. Transfection of 100 nM of siRNA were carried out using Interferin (PolyPlus Transfection), and the cells were analysed at least 72 h after transfection.

HeLa cells were transfected with WT or D50A CMG2 tagged with the V5 epitope using Fugene according to the manufacturer's protocol (Promega). After 48 h, confluent cells were detached with PBS–EDTA, counted and equally distributed on coated dishes. A total amount of 5 µg of collagen I, collagen IV, collagen VI, laminin, fibronectin and PA were coated on a six-wells plate for 2 h at room temperature. Cell adhesion was allowed for 60 min at 37 °C, and the cells were subsequently washed 3 × in PBS 1 × at 4 °C and lysed for 30 min at 4 °C in IP buffer (0.5% NP-40, 500 mM Tris-HCl pH 7.4, 20 mM EDTA, 10 mM NaF, 30 mM sodium pyrophosphate decahydrate, 2 mM benzamidin, 1 mM PMSF, 1 mM N-ethyl maleimide), supplemented with a cocktail of proteases inhibitors (Roche) and phosphatase inhibitors (Sigma). Cell lysates were spun down at 5,000 r.p.m. for 5 min and the supernatant were submitted to immunoprecipitation using 1 µl of mouse anti-V5 antibody and protein G beads overnight at 4 °C on a rotating wheel. Protein G beads were then washed 3 × and boiled in Laemmli buffer under reducing conditions for 5 min before analysis by SDS–PAGE using 4–12% Bis-Tris gradient gels under reducing condition and western blotting with phospho-Tyr antibody and rabbit anti-V5 antibody.

**Co-immunoprecipitation.** Confluent RPE1 cells were detached with PBS–EDTA, counted and equally distributed on coated dishes. A total amount of 5 µg of collagen I, collagen IV, collagen VI, laminin, fibronectin and PA were coated on a six-wells plate for 2 h at room temperature. Cell adhesion was allowed for 60 min at 37 °C, and the cells were subsequently washed 3 × in PBS 1 × at 4 °C and lysed for 30 min at 4 °C in CoIP buffer (PBS 1 ×, 1% Triton X-100 and protease inhibitor cocktail (Roche)) for 30 min on ice. Lysates were then spin down at 5,000 r.p.m. on a tabletop centrifuge and the supernatant incubated with protein G beads for 1 h on a wheel at 4 °C. The beads supernatant was then incubated with 3 µl of rabbit anti-β-arrestin for 3 h on a wheel at 4 °C, to be finally captured on 40 µl of protein G beads for 1 h on a wheel at 4 °C. Immunoprecipitates were then washed 3 × in CoIP buffer and samples were boiled in non-reducing Laemmli buffer for 5 min before SDS–PAGE and western blotting against endogenous CMG2.

**Endocytosis assay.** Control or RPE1 cells stably expressing a small hairpin RNA targeting CMG2 mRNA were seeded on plates and incubated overnight in DMEM complemented with Gropro (Zen-Bio), to remove the collagen VI present in FBS. The next days, cells were incubated or not for 60 min with 100 nM bafilomycin (Sigma). Thirty minutes before the addition of purified collagen VI tetramer (kind gift from P. Bonaldo), 5% BSA was added to the cell culture medium to block aspecific binding. A measure of 1 µg ml$^{-1}$ of purified collagen VI tetramer was then added to cells. Cells were lysed in IP buffer (0.5% NP-40, 500 mM Tris-HCl pH 7.4, 20 mM EDTA, 10 mM NaF, 30 mM sodium pyrophosphate decahydrate, 2 mM benzamidin, 1 mM PMSF, 1 mM N-ethyl maleimide, cocktail of proteases inhibitors (Roche) and phosphatase inhibitors (Sigma)) 1, 3 and 6 h after collagen VI addition. Protein quantity was measured using BCA assay kit (Pierce) and equal amount of proteins boiled in Laemmli buffer under non-reducing conditions for

5 min before analysis by SDS–PAGE using 4–12% Bis-Tris gradient gels and western blotting with collagen VI antibody and rat monoclonal anti-CMG2 antibody. Actin was used as a loading control.

**Gelatin zymography.** Cells were put in serum-free media the night before the experiment. Cell supernatant (SN) were normalized according to protein concentration. Non-reducing Laemmli buffer was added to SN and let rest for 10 min at room temperature. Samples were then loaded on a 0.1% gelatin acrylamide gel and run at 125 V. The gel was then incubated 30 min at room temperature in renaturing buffer (2.5% Triton X-100 in double distilled H$_2$O—ddH$_2$0). Renaturing buffer was then replaced by developing buffer for 30 min to equilibrate (50 mM Tris-HCl pH 8, 5 mM CaCl$_2$, 0.2% azide). Fresh developing buffer was then added and the gel was incubated at 37 °C overnight. The gel was then washed 10 min with ddH$_2$O, and put in fixation solution (40% methanol, 10% acetic acid, ddH$_2$O) for 20 min at room temperature. After fixation, the gel was washed in ddH$_2$O and stained with Coomassie blue for at least 1 h at room temperature. The gel was finally destained in ddH$_2$O overnight at 4 °C without shaking and analysed the next day.

**RNA extraction and quantitative real-time PCR.** Total RNA was extracted from confluent cell culture using RNA easy mini extraction kit (Qiagen). Mice tissues were disrupted using Lysing Matrix tube (MP Biomedicals) and homogenized in a tissue lyser (Qiagen) 3 × 5 min, with 2 min resting period at 4 °C between each round, followed by RNA extraction using RNA easy mini extraction kit (Qiagen). RNA concentration was measured by spectrometry and 500 ng of total RNA was used for reverse transcription. We then used a 1/5 dilution of the cDNA to perform quantitative real-time PCR using SYBR green MasterMix (Life Technology) on a 7900HT Real-Time PCR System (Applied Biosystem). mRNA level was normalized using TATA-binding protein and β-microglobulin for RNA derived from human cell culture and cytochrome *c* oxydase and β-actin for RNA derived from mice tissues. See Supplementary Table 3 for the list of the primers used in this study.

**Statistical analysis.** Calculations were performed using GraphPad Prism 5.0 (GraphPad Software, Inc.). When two groups were compared, an unpaired Student's *t*-test was used to test significance, variance between the groups were of similar value. Statistical significance was reached for *P*-value < 0.05.

**Data availability.** The MS proteomics data have been deposited to the ProteomeXchange Consortium via the PRIDE[42] partner repository with the data set identifier PXD006268 and 10.6019/PXD006268.

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

## Acknowledgements

We thank patients and parents with their courage, help and support. We are grateful to R. Wagener for sharing valuable antibodies against the different chains of collagen VI. We thank D. Trono and D. Duboule for critical reading of the manuscript. We thank A. Boemio for the help and dedication. We thank G. Knott and S. Rosset for their work and expertise on EM. We thank R. Hamelin and other members of the EPFL Proteomics Core Facility for mass spectrometry analysis, and the EPFL Histology core facility with their support in tissue analysis. We thank all the members of the F.G.v.d.G.'s lab for their discussions and suggestions. This work was supported by the Swiss National Science Foundation, the Gelu Foundation, the Francis and Marie-France Minkoff Foundation, the Solis Foundation and the Associazione ISI.

## Author contributions

J.B., B.K., J.D. and L.A. designed and performed experiments. A.P. provided pathological expertize. A.S.B., S.U. and A.S.-F. provided medical expertize and assistance with HFS patients. E.L. provided medical expertize and medical assessment of the HFS patient involved in this study. P.B. provided the *Col6a1*$^{-/-}$ mice and scientific expertize. J.B. and F.G.v.d.G. wrote the manuscript. All authors were involved in scientific discussion.

## Additional information

**Competing interests:** The authors declare no competing financial interests.

