## [Peer Review File · Nature Communications]

Reviewers' comments:

Reviewer #1 (Remarks to the Author):

In this paper the authors attempt to provide a mechanism whereby decreased expression of GMC2 leads to accumulation of collagen VI, fibrosis (particularly in the uterus) and ultimately sterility.

The paper presents major flaws that overall dampen the enthusiasm for the study presented.

At present the data presented are scant and of descriptive nature. This type of qualitative and superficial work need to be significantly improved for the quality of this journal.

The number of mice used seems quite low (with n=3)

Although the authors might provide evidence that GMC2 might bind collagen VI they fail to explain how exactly GMC2/collagen VI interaction prevents collagen VI accumulation.

At present there is no convincing evidence that loss of GMC2 directly causes accumulation of collagen VI as a mechanism is not provided.

the rationale for looking at the possibility that GMC2 might bind collagen VI is not entirely clear.

The fact that crossing the GMC2KO mice with collagen VIKO mice is interesting but again does not explain how and importantly if GMC2 is indeed involved in collagen VI accumulation.

Reviewer #2 (Remarks to the Author):

The manuscript by Bürgi et al describes the role of the extracellular matrix protein collagen VI in the pathogenesis of the Hyaline Fibromatosis Syndrome (HFS) caused by loss-of-function mutations in Capillary Morphogenesis Gene 2 (CMG2), a.k.a. Anthrax Toxin Receptor 2 (ANTXR2) gene. HFS is a painful and disfiguring disease characterized by the progressive growth of subcutaneous nodules. This gene encodes the transmembrane surface protein CMG2 that contains an extracellular von Willebrand A (vWA) domain known to mediate protein-protein interactions.

In the present study, the authors showed that CMG2 loss-of-function in HFS patients and in null mice is associated with fibrotic deposition of the extracellular protein collagen VI in tissues (skin and uterus, respectively) with no change in collagen VI genes expression. They showed that collagen VI binds to CMG2 through its extracellular von Willebrand A (vWA) domain and triggers tyrosine phosphorylation. Interestingly, the authors further showed that the fibrosis phenotype is rescued by crossing *Antxr2*^{-/-} with

Col6a1^{-/-} mice. They concluded that CMG2 acts as a receptor for collagen VI and that the amount of collagen VI in tissues is regulated through its binding to the transmembrane protein CGM2.

The data are interesting but not entirely novel since CMG2 was already shown to function as a collagen receptor that is essential for maintaining collagen homeostasis in the uterus in mice (Peters et al, 2012). It has been reported that CMG2 is endocytosed by cells and transported to the late endocytic pathway upon binding of its ligand anthrax protective antigen. Consistently, the underlying mechanism proposed by the authors is that upon its binding to CGM2, collagen VI is internalized and transported to the late endocytic pathway where it is degraded, preventing deleterious collagen VI accumulation in tissues (l163-165). It is regrettable that this mechanism is not shown experimentally at least by co-staining with antibodies against collagen VI and endocytic markers in cells or by blocking collagen VI endocytosis. It will add value to the manuscript.

Major points

1. As said above, the fact that CMG2 acts as an ECM receptor is not new. CMG2 was shown to bind collagen IV, fibronectin and laminin by solid phase assays (Bell et al, 2001). The data presented in this paper are somewhat controversial. Using the same method, CGM2 was shown to bind to collagen VI but not to collagen IV, collagen I, fibronectin and laminin. These controversial data are not further discussed.
2. The analysis of the protein amount is difficult if several unidentified bands appear on the blot. Please indicate which band corresponds to which protein or collagen chains in Figure 1c and Suppl Fig 2 d and e. Did the multiple bands detected by western blot correspond to protein breakdown or a lack of antibody specificity? Collagen VI and IV are heterotrimeric molecules, do the antibodies used in this study recognize the different collagen α -chains?
3. Collagen VI is an extracellular matrix protein with broad distribution in several tissues and CMG2 is also widely expressed in human and murine tissues. It is not clear why loss of function of CMG2 provokes only defects in uterus in female mice and formation of subcutaneous nodules in HSF patients.
4. The composition of the 100kDa band shown in Fig1b was shown to contain the three chains of collagen VI and the small proteoglycan decorin although their size in kDa is very different (from 40 to 344 kDa). Could you clarify this point?
5. Decorin is present in the major 100kDa band indicating that it also accumulates in nodules identified by mass spectrometry (Fig 1b) but is not present in the list of most enriched proteins in HFS patients (Table 1). Could you explain this apparent discrepancy? Collagen VI binds to decorin: could CGM2 internalize the complex?
6. Collagen VI forms filaments that are easily identified by TEM. It could be instructive to analyze ultrastructural changes of the fibrotic extracellular matrix of the mutant uterus by TEM.
7. Fibrotic extracellular matrix activates a profibrotic positive feedback loop. TGF- β is the major pro-fibrotic growth factor. Does TGF- β expression increase in Antxr2^{-/-} mice or in patients with HSF?

Minor points:

l136: Change figure 3c to Figure 3b in the text.

l142: Col6a3 mRNA is the only transcript that is not shown in Figure 3d - Please correct the text. Actually, the results for the 3 chains should be shown as they assemble together to form the collagen VI molecule.

Is Supp Figure 2 reproduced with permission?

What is the faint band observed in Figure 2c Antrx2-/-?
Fig 2b: code color is missing.

Reviewer #3 (Remarks to the Author):

The authors report on an unexpected function of the protein encoded by CMG2, suggesting that it acts also as a receptor for collagen VI and that in patients with hyaline fibromatosis, loss of CMG2 function is associated with a large build-up of collagen VI in nodules of affected patients. They found that in Antxr2 35 -/- mice, build-up of collagen VI protein, but without any real change of collagen VI gene expression, is associated with progressive uterine fibrosis and female sterility.

When the authors crossed Antxr2 37 -/- with Col6a1 38 -/- mice, uterine structure and recovery of female fertility was obtained.

The authors conclude that without CMG2 function, build-up of collagen VI occurs in the extracellular matrix, which in turn causes loss of normal tissue function and disease. The authors suggest that CMG2 is a collagen VI receptor and maybe important in understanding other diseases.

The results provided here are generally consistent with the hypothesis that CMG2 acts as a receptor for collagen VI but if this notion is to form a central component of the argument of the paper, there are missing data elements that would be required to make a more persuasive case for this supposition.

1) if CMG2 acts as an endocytic or phagocytic receptor, then the authors should show in cultured cells that express (or not) CMG2, when cultured on type VI collagen or fibronectin or type I collagen, then the expression of this putative receptor is associated with measurable (and indeed large-scale) increases of internalized collagen that enters into the lysosomal pathway of degradation. These findings would help to support their finding of the excessive extracellular abundance of type VI collagen in the absence of the receptor in the lesions of interest.

2) Their worthwhile finding that type VI collagen enables activation of Src-related adhesion signaling is good but these data are not accompanied by sufficient experimental details or controls, namely that collagens and other glycoproteins in the pericellular matrix do not enable receptor activation through other matrix molecules in the glycocalyx. One approach here would be to carefully strip the glycocalyx from cultured cells prior to conduct of these assays.

3) The loss of CMG2 expression may affect other important pathways that are associated with degradation of collagen VI, specifically the expression of lysosomal hydrolases (e.g. cathepsins) and the activation of Rabs that are important for intracellular trafficking in the degradation pathway. The determination of whether the expression of these molecules and intracellular degradation pathways is influenced by CMG2 should be provided.

4) The loss of CMG2 expression does not seem to be associated with gene expression of collagen VI, suggesting that it is indeed the degradation side of the remodeling equation that leads to the disorder of interest. Since it is thought that much of matrix remodeling is associated with pericellular proteolysis, the authors must provide an account of the expression and relative proteolytic activities of the matrix metalloproteinases that may mediate degradation of type VI collagen in pericellular environments.

5) A surprising finding in the data set provided here is the remarkable abundance (indeed predominance) of type VI collagen over type I and type III collagens in the extracellular matrix. This unusual shift of the relative abundance of these collagens in these lesions suggests that collagens type I and type III are normally or at least relatively normally, remodeled in the absence of the CMG2 receptor. Since the relative abundance of type I collagen to type VI collagen is normally many fold higher in skin, this would suggest that other systems that mediate remodeling of type VI collagen are almost completely shut down and that the remodeling of type I and type III collagens are largely unaltered when CMG2 is not expressed. If this were the case, then the binding data provided in the data set here would be expected to show much more preferential differences in binding between collagen types than are depicted. In other words, the binding of the collagen types to the putative receptor and the abundance of the extracellular matrix collagens and their stoichiometry don't seem to really add up.

6) If CMG2 is indeed going to be put forward as a specific, collagen VI receptor, then more convincing and detailed experiments are needed to show receptor binding and specificity than are provided here. Connected to this idea is the need to provide structural data and binding motifs by which CMG2 could provide high affinity binding to discrete domains of collagen VI (but not other collagen types).

We thank the reviewers for their comments, which have allowed us to improve the manuscript.

Response to Reviewer 1:

At present the data presented are scant and of descriptive nature. This type of qualitative and superficial work need to be significantly improved for the quality of this journal.

We have added significant additional data to the manuscript and in particular provide more mechanistic information. We show that binding of collagen VI leads to phosphorylation of CMG2 and recruitment of Beta-arrestin, which is required for CMG2 ubiquitination and endocytosis. We moreover show that CMG2 mediates the lysosomal degradation of Collagen VI, providing an explanation as to why CMG2 loss-of-function or knock out leads to collagen VI accumulation in HFS patient nodules and *Antxr2*^{-/-} mice uteri.

The number of mice used seems quite low (with n=3).

The uterine phenotype was observed in at least 19 *Antxr2*^{-/-} female mice, as now mentioned. Also the rescue of the parturition defect in *Antxr2*^{-/-}*Col6a1*^{-/-} mice was observed in 6 animals, with multiple pregnancies. For RT-qPCR experiments, we have now increased the number of mice to 7, confirming the previous observations.

- Although the authors might provide evidence that CMG2 might bind collagen VI they fail to explain how exactly CMG2/collagen VI interaction prevents collagen VI accumulation.

As mentioned above, we now show, in figure 5, that purified full length Collagen VI when added to RPE1 cells undergoes degradation in a CMG2 dependent manner. This degradation was inhibited by the lysosomal inhibitor Bafilomycin A, indicating that collagen VI undergoes lysosomal degradation. Together with the evidence that collagen VI can bind to CMG2, triggers phosphorylation and the recruitment of β -arrestin, this additional data show that CMG2 is a receptor for Collagen VI, that it can mediate its cellular uptake and degradation in lysosomes.

Response to Reviewer 2:

- the underlying mechanism proposed by the authors is that upon its binding to CMG2, collagen VI is internalized and transported to the late endocytic pathway where it is degraded, preventing deleterious collagen VI accumulation in tissues (163-165). It is regrettable that this mechanism is not shown experimentally at least by co-staining with antibodies against collagen VI and endocytic markers in cells or by blocking collagen VI endocytosis. It will add value to the manuscript.

We now show additional evidence in figure 5 in support of this model. In addition to the collagen VI-induced phosphorylation of CMG2, we show that Collagen VI triggers the recruitment of Beta-arrestin to the cytoplasmic tail of CMG2. Beta-arrestin is an adapter protein required for the anthrax toxin induced ubiquitination and endocytosis of CMG2. More importantly, as mentioned in the response to reviewer 1, we now also show that purified full length Collagen VI when added to RPE1 cells undergoes degradation in a CMG2-dependent manner. This degradation was inhibited by the lysosomal inhibitor Bafilomycin A, indicating that collagen VI undergoes lysosomal degradation. Together with the evidence that collagen VI can bind to CMG2, triggers phosphorylation and the recruitment of β -arrestin, this additional data show that CMG2 is a receptor for Collagen VI, that it can mediate its cellular uptake and degradation in lysosomes.

- As said above, the fact that CMG2 acts as an ECM receptor is not new. CMG2 was shown to bind collagen IV, fibronectin and laminin by solid phase assays (Bell et al, 2001). The data presented in this paper are somewhat controversial. Using the same method, CGM2 was shown to bind to collagen VI but not to collagen IV, collagen I, fibronectin and laminin. These controversial data are not further discussed.

As mentioned by the reviewer, in 2001, Bell et al. proposed that the vWA of CMG2 can bind to commercially available Collagen IV, Laminin and fibronectin. The experimental approach that was used was to coat ELISA plates with recombinant CMG2 vWA domain and to monitor the binding of biotinylated commercially available ECM proteins. The biotinylation efficiency of the different ECM proteins was however not determined, and may well vary significantly between different collagens, laminin and fibronectin, thus influencing the apparent binding revealed using streptavidin. We therefore choose to coat the ELISA plates with the various ECM proteins and to monitor binding of GST-tagged CMG2 vWA using an anti-GST antibody, to have the same readout for all conditions. Using this approach, binding to collagen VI was more pronounced than to collagen IV, laminin and fibronectin.

To explain the difference with the previous report we now mention line 193, page 8: *The discrepancy between the two findings might be due to the difference in readout of the ELISA assays and the fact that ECM proteins were coated onto the ELISA plates in our approach as opposed to the vWA domain in the previous study.*

Also Bell et al. analyzed in vitro binding of recombinant CMG2 vWA domain but did not address whether full length CMG2 was an ECM receptor. We show that, in the cellular context, full length CMG2 binding to collagen VI leads to CMG2 phosphorylation, β -arrestin recruitment and ligand degradation. This does not exclude that CMG2 could have a lower affinity for other ECM proteins that would not lead to intracellular events.

The analysis of the protein amount is difficult if several unidentified bands appear on the blot. Please indicate which band corresponds to which protein or collagen chains in Figure 1c and Suppl Fig 2 d and e. Did the multiple bands detected by western blot correspond to protein breakdown or a lack of antibody specificity? Collagen VI and IV are heterotrimeric molecules, do the antibodies used in this study recognize the different collagen α -chains?

Mature Collagen VI is indeed a trimer of α 1, 2, 3 Collagen VI proteins. This trimer then assembles into a dimer of trimers, and then a tetramer of trimers. Assembly into these higher order structures occurs intracellularly prior to secretion.

Unless specified, we performed electrophoresis under reducing conditions and probed our western blots with polyclonal antibodies (for example abcam ab6588 for collagen VI). Thus the different α chains, as well as degradation products thereof, can potentially be observed. α 1 and 2 Collagen VI are about 100kDa and α 3 Collagen VI is about 345kDa. There thus appears to be degradation in the patient nodules, but we can not determine, as mentioned in the text, whether degradation occurs in the nodule itself or in the biopsy which could not be analyzed at the site of surgery.

For the analysis of the mouse tissues, we have now added in Fig. 2 an analysis using monoclonal antibodies targeted against the different α chains of collagen VI, with a quantification in Suppl. Fig. 3d.

Collagen VI is an extracellular matrix protein with broad distribution in several tissues and CMG2 is also widely expressed in human and murine tissues. It is not clear why loss of function of CMG2 provokes only defects in uterus in female mice and formation of subcutaneous nodules in HSF patients.

This is indeed an intriguing question that will require further investigation. We do observe accumulation of Collagen VI other tissue for example the Gut of aged CMG2 knock-out mice. Thus Collagen VI accumulation can occur in other organs, but to a lesser extent.

We now mention in the text that collagen VI has been reported to increase in the mouse uterus during pregnancy and subsequently returns to basal levels. Thus proper degradation of collagen VI might be particularly important in the uterus.

The composition of the 100kDa band shown in Fig1b was shown to contain the three chains of collagen VI and the small proteoglycan decorin although their size in kDa is very different (from 40 to 344 kDa). Could you clarify this point?

Decorin is present in the major 100kDa band indicating that it also accumulates in nodules identified by mass spectrometry (Fig 1b) but is not present in the list of most enriched proteins in HFS patients (Table 1). Could you explain this apparent discrepancy? Collagen VI binds to decorin: could CGM2 internalize the complex?

As was shown in the table of Figure 1, Decorin is detected by mass spec in the nodules of HFS patient when analyzing the prominent 100kDa band. Only 2 peptides were however detected in contrast to 48 to 109 for the collagen VI chains. We analyzed the detection of decorin in more detail. The proteomic analysis of the patient tissue lysates in fact shows that decorin is actually less present in nodules than in non-nodular tissues (see figure below). To avoid misleading the reader we have removed decorin from the table and indicated in the legend that only proteins for which >2 peptides were detected are shown.

Regarding the first point on the molecular weight of decorin and collagen VI chains: small molecular weight proteins migrate along the entire length of the lane and traces remain, therefore it is not uncommon, especially with the very sensitive modern mass spectrometers, to detect proteins that have a lower molecular weight than that corresponding to excised band.

The question of the review has led us to analyze the peptides detected by mass spectrometry in more detail. We found that the Col6a3 peptides mostly map to a central part

of the col6A3 polypeptide chain suggesting cleavage of the protein has occurred (Suppl. Fig. 2). Interestingly, when analyzing the MS data obtained from different bands of the head nodule sample (higher and lower than the 100 kDa band, suppl. fig.2), we found that the higher the molecular weight of the band on the gel, the wider the protein coverage was. Thus collagen VI that accumulates in nodules might be partially processed in the nodules and/or the biopsy, as now mentioned above and in the text.

- Collagen VI forms filaments that are easily identified by TEM. It could be instructive to analyze ultrastructural changes of the fibrotic extracellular matrix of the mutant uterus by TEM.

We have now analyzed both the patient head nodule and the uterus from *Antxr2^{-/-}* mice by EM (Figs. 1 and 2). Both reveal accumulation of non-fibrillar material and loss of cellularity. The dark granular material seen in the uterus could be microbeaded filaments of Collagen VI.

- Fibrotic extracellular matrix activates a profibrotic positive feedback loop. TGF-beta is the major pro-fibrotic growth factor. Does TGF-beta expression increase in *Antxr2^{-/-}* mice or in patients with HSF?

By QRT-PCR, we observed a small but significant increase in TGF-Beta1 mRNA in uteri of *Antxr2^{-/-}* mice. However, no corresponding increase in TGF-Beta1 target genes was observed. Thus collagen VI accumulation is unlikely to be initiated by activation of the TGF β pathway. This new data is now described in Fig. 2e and the text.

Minor points:

- I136: Change figure 3c to Figure 3b in the text.

This was corrected.

- I142: Col6a3 mRNA is the only transcript that is not shown in Figure 3d - Please correct the text. Actually, the results for the 3 chains should be shown as they assemble together to form the collagen VI molecule.

We added Col6a3 mRNA profile in HFS patient fibroblasts.

- Is Supp Figure 2 reproduced with permission?

Suppl Figure 2 was generated from publically available coordinates (PDB ID: 1SHT) using Chimera as now specified.

- What is the faint band observed in Figure 2c *Antrx2^{-/-}*?

We suppose the reviewer refers to the previous figure 3c. We are not sure which band she/he is referring to. In any case as mentioned above, we have now replaced this figure with a western blot analysis using monoclonal antibodies against the different collagen VI alpha chains.

- Fig 2b: code color is missing.

The color code has been added.

Response to Reviewer 3:

If CMG2 acts as an endocytic or phagocytic receptor, then the authors should show in cultured cells that express (or not) CMG2, when cultured on type VI collagen or fibronectin or type I collagen, then the expression of this putative receptor is associated with measurable (and indeed large-scale) increases of internalized collagen that enters into the lysosomal pathway of degradation. These findings would help to support their finding of the excessive extracellular abundance of type VI collagen in the absence of the receptor in the lesions of interest.

As mentioned to reviewers 1 and 2, we now show new evidence in support of this model (Fig. 5). In addition to the collagen VI-induced phosphorylation of CMG2, we now show that Collagen VI triggers the recruitment of Beta-arrestin to the cytoplasmic tail of CMG2. Beta-arrestin is an adapter protein required for the anthrax toxin induced ubiquitination and endocytosis of CMG2. More importantly, we now also show that purified full length Collagen VI when added to RPE1 cells undergoes degradation, in a CMG2-dependent manner. This degradation was inhibited by the lysosomal inhibitor Bafilomycin A, indicating that collagen VI undergoes lysosomal degradation. Together with the evidence that collagen VI can bind to CMG2, triggers phosphorylation and the recruitment of β -arrestin, this additional data show that CMG2 is a receptor for Collagen VI, that it can mediate its cellular uptake and degradation in lysosomes. These findings are consistent with the reported insensitivity of Collagen VI to extracellular metalloproteinases, and its sensitivity to serine proteases, which are the enzymes found in lysosomes.

Their worthwhile finding that type VI collagen enables activation of Src-related adhesion signaling is good but these data are not accompanied by sufficient experimental details or controls, namely that collagens and other glycoproteins in the pericellular matrix do not enable receptor activation through other matrix molecules in the glycocalyx. One approach here would be to carefully strip the glycocalyx from cultured cells prior to conduct of these assays.

We have now clarified the protocol used in these experiments in the legend of the corresponding figure. Different ECM proteins were coated onto tissue culture dishes. Cells from the same culture, thus with the same pericellular matrix, were added to the various dishes, excluding a role of the pericellular matrix in the src activation which was only observed upon addition of the cells to collagen VI coated dishes.

The loss of CMG2 expression may affect other important pathways that are associated with degradation of collagen VI, specifically the expression of lysosomal hydrolases (e.g. cathepsins) and the activation of Rabs that are important for intracellular trafficking in the degradation pathway. The determination of whether the expression of these molecules and intracellular degradation pathways is influenced by CMG2 should be provided.

To investigate whether CMG2 loss affect intracellular degradation pathways, we monitored uptake and degradation of mouse IgG molecules. IgG degradation was unaffected by CMG2 knock-down (shown in Suppl. Figure 3). This observation is consistent with our findings that silencing of CMG2 in RPE1 cells does not prevent endocytosis and intracellular trafficking of the anthrax toxin via its other receptor TEM8.

The loss of CMG2 expression does not seem to be associated with gene expression of collagen VI, suggesting that it is indeed the degradation side of the remodeling equation that leads to the disorder of interest. Since it is thought that much of matrix remodeling is associated with pericellular proteolysis, the authors must provide an account of the expression and relative proteolytic activities of the matrix metalloproteinases that may mediate degradation of type VI collagen in pericellular environments.

We have added a new section on the activity of MT1MMP/MMP14 and MMP2 in nodules, mouse uteri and cells (Fig. 3). We analyzed the nodules of the HFS patient and saw a decrease in MT1MMP activation and in active MMP2. This is not a direct effect of the CMG2 loss-of-function since we could not detect differences in MMP2 and 14 activity in a patient derived cells. It is known that fibrosis leads to capture and accumulation of inactive MMP2, which is a positive feedback loop for the fibrotic phenotype. Thus in HFS nodules, the massive accumulation of collagen VI could be due to the combined effect of 1) decrease of collagen VI uptake and degradation by cells; 2) a decrease in metalloprotease activity, which could progressively change in the overall architecture of the ECM. In mouse uteri we however found the opposite, an increase in MT1MMP and MMP2 active form. Finally, we addressed the issue in a more controlled tissue culture system. Silencing of CMG2 in RPE1 cells led to an activation of MT1MMP and MMP2. All together, these observations indicate that changes in metalloprotease activity cannot account for the accumulation of collagen VI.

A surprising finding in the data set provided here is the remarkable abundance (indeed predominance) of type VI collagen over type I and type III collagens in the extracellular matrix. This unusual shift of the relative abundance of these collagens in these lesions suggests that collagens type I and type III are normally or at least relatively normally, remodeled in the absence of the CMG2 receptor. Since the relative abundance of type I collagen to type VI collagen is normally many fold higher in skin, this would suggest that other systems that mediate remodeling of type VI collagen are almost completely shut down and that the remodeling of type I and type III collagens are largely unaltered when CMG2 is not expressed. If this were the case, then the binding data provided in the data set here would be expected to show much more preferential differences in binding between collagen types than are depicted. In other words, the binding of the collagen types to the putative receptor and the abundance of the extracellular matrix collagens and their stoichiometry don't seem to really add up.

As we now mention in the manuscript, as opposed to other collagens, collagen VI was reported to be insensitive to metalloproteases, but sensitive to serine hydrolases, consistent with the lysosomal degradation proposed here. Also the analysis of the mouse uteri indicates that metalloproteases are still active, potentially even more active, explaining why other collagens do not appear to accumulate. The activity of the metalloproteases is however likely to vary considerably with the age of the fibrotic tissue. We only have access to "old" nodules, i.e. when they are so big that they need to be removed to improve the patients quality of life. The massive accumulation of matrix in these nodules is likely to trap various molecules and affect their structure and/or activity.

If CMG2 is indeed going to be put forward as a specific, collagen VI receptor, then more convincing and detailed experiments are needed to show receptor binding and specificity than are provided here. Connected to this idea is the need to provide structural data and binding motifs by which CMG2 could provide high affinity binding to discrete domains of collagen VI (but not other collagen types).

We have strengthened the evidence that CMG2 is a receptor for collagen VI by additionally showing the collagen VI-induced recruitment of β -arrestin to CMG2 and the CMG2-dependent lysosomal degradation of collagen VI.

We believe that the precise structural characterization of the CMG2-collagen VI interaction goes beyond the scope of this manuscript. As we now mention in the text, we already have information regarding the domains in both proteins that are involved in the interaction: on the

CMG2 side, it is the MIDAS domain of the vWA domain and on the collagen VI side it is the triple helical domain, since this is the domain present in the commercially available collagen VI which is obtained by pepsin degradation of tissues.

Affinity of CMG2 for the collagen VI is a very interesting question that we will address in the future. This affinity could well be fairly low. This is suggested by the fact that addition of anthrax toxin leads to the displacement of the endogenous ligand. Low affinity would allow CMG2 to control the level of its ligand in the extracellular space, without depleting collagen VI from the ECM.

Reviewers' comments:

Reviewer #1 (Remarks to the Author):

The authors have addressed most of the critiques previously raised and they provide convincing argument that the Anthrax receptor plays an anti-fibrotic role by favoring collagen IV uptake and most likely degradation.

There are however some concerns still pending:

1) The authors state that they have analyzed up to 7 mice with identical and/or similar phenotype. Yet figure 1 legend still states that n = 3 were analyzed. The WB showing that in the Anthrax ko mice there is upregulation of collagen IV is not quantified and more than one mouse needs to be shown, ideally 3 mice should be shown and quantification of the now n=7 needs to be provided.

2) The same argument is valid for figure 4 the crossing of the anthrax ko mice with collagen VI mice. The figure is of descriptive and qualitative nature. Given that this is a major finding of the paper, a better quantification/evaluation of this figure is needed.

Reviewer #2 (Remarks to the Author):

The authors have addressed all the points raised by the reviewers. However, I have concerns about the interpretation of the TEM images of the extracellular matrix of the Antxr2-/- uterus (Figure 2b). I am not at all convinced that the dark granular material seen in the ECM of uterus at high magnification is Collagen VI beaded filaments. The authors should show higher magnification of a small area of the image shown in the upper panel to facilitate interpretation.

Reviewer #3 (Remarks to the Author):

The authors have done a generally conscientious job in addressing the list of reviewers' comments, which is always for authors, a tedious and somewhat trying business. Nevertheless, the intention of the reviewers is to help the authors improve the quality of the manuscript, which is also first and foremost in the mind of the editor(s) of the journal. In this context, my overall reading of the responses of the authors to the reviewers' comments was very positive with the exception of the following two points made by me earlier, which I don't think have been addressed adequately with the needed experiments:

1) In the previous review I indicated that "Since the relative ratio of abundance of type I collagen to type VI collagen is normally many fold higher in skin, this would suggest that other systems that mediate remodeling of type VI collagen are almost completely shut down and that the remodeling of type I and type III collagens are largely unaltered when CMG2 is not expressed. If this were the case, then the binding data provided in the data set here would be expected to show much more preferential differences in binding between collagen types than are depicted. In other words, the binding of the collagen types to the putative receptor and the abundance of the extracellular matrix collagens and their stoichiometry don't seem to really add up."

The authors respond by indicating that: As we now mention in the manuscript, as opposed to other collagens, collagen VI was reported to be insensitive to metalloproteases, but sensitive to serine hydrolases, consistent with the lysosomal degradation proposed here."

This is an idea that is not supported by other work and the authors themselves have not provided evidence to support this. Data are needed to underpin this remarkable assertion.

The authors go on to say: "Also the analysis of the mouse uteri indicates that metalloproteases are still active, potentially even more active, explaining why other collagens do not appear to accumulate."

What is the evidence for this statement?

Later in the same part of the response, the authors say:

"The activity of the metalloproteases is however likely to vary considerably with the age of the fibrotic tissue. We only have access to "old" nodules, i.e. when they are so big that they need to be removed to improve the patients quality of life. The massive accumulation of matrix in these nodules is likely to trap various molecules and affect their structure and/or activity."

The notion that the the nodules may trap other molecules and, in some way, hinder remodeling processes is again, a remarkable and potentially interesting assertion. What is the evidence for this?

2) In the previous review, I indicate that: "If CMG2 is indeed going to be put forward as a specific, collagen VI receptor, then more convincing and detailed experiments are needed to show receptor binding and specificity than are provided here. Connected to this idea is the need to provide structural data and binding motifs by which CMG2 could provide high affinity binding to discrete domains of collagen VI (but not other collagen types)."

No substantive data have been provided that address these points, and this is, I believe, fundamental to the mechanism which they are proposing. This should be addressed with new data.

Reviewer #1

- 1) *The authors state that they have analyzed up to 7 mice with identical and/or similar phenotype. Yet figure 1 legend still states that n = 3 were analyzed. The WB showing that in the Anthrax ko mice there is upregulation of collagen IV is not quantified and more than one mouse needs to be shown, ideally 3 mice should be show and quantification of the now n=7 needs to be provided.*

We have corrected the legend, indicating that n=7.

Regarding the WB of the mouse uteri, we now show the blots for the different chains on 4 mice. We thank the reviewer for this suggestion since it allows to illustrate that while collagen VI always increases, there is a variation between chains and between mice.

We however choose not to show a quantification of the increases, as mentioned in the legend of figure 2. Indeed Western blot analysis has a rather small dynamic range. The increases in collagen VI alpha chains that we observe are in many cases outside of the dynamic range of western blotting and thus cannot be quantified properly.

- 2) *The same argument is valid for figure 4 the crossing of the anthrax ko mice with collagen VI mice. The figure is of descriptive and qualitative nature. Given that this is a major finding of the paper, a better quantification/evaluation of this figure is needed.*

We agree that figure 4 showing the histology of the uterus in single and double KO mice is descriptive. The major finding of the paper is however that the double KO mice, as opposed to the single *Antrx2* KO mice, are fertile. The quantification was thus performed on the fertility, where 6 out of 6 mice gave birth and underwent multiple pregnancies, whereas not a single *Antrx2* KO mouse ever gave birth as also observed in two other studies by 2 different labs.

Also it is not clear what one would quantify in the histology given that the uterus architecture is either normal or completely disrupted.

We now show, in the supplementary figures, H&E staining on the uteri of 2 additional *Antrx2* KO mice and 2 double KO mice to illustrate that the recovery of the architecture is a reproducible feature that accompanies the recovery of fertility.

Reviewer #2

*The authors have addressed all the points raised by the reviewers. However, I have concerns about the interpretation of the TEM images of the extracellular matrix of the *Antxr2*^{-/-} uterus (Figure 2b). I am not at all convinced that the dark granular material seen in the ECM of uterus at high magnification is Collagen VI beaded filaments. The authors should show higher magnification of a small area of the image shown in the upper panel to facilitate interpretation.*

Rather than showing higher magnification, we have now performed anti-collagen VI immunostaining which shows striking collagen VI staining in the extracellular environment.

Reviewer #3:

1) In the previous review I indicated that "Since the relative ratio of abundance of type I collagen to type VI collagen is normally many fold higher in skin, this would suggest that other systems that mediate remodeling of type VI collagen are almost completely shut down and that the remodeling of type I and type III collagens are largely unaltered when CMG2 is not expressed. If this were the case, then the binding data provided in the data set here would be expected to show much more preferential differences in binding between collagen types than are depicted. In other words, the binding of the collagen types to the putative receptor and the abundance of the extracellular matrix collagens and their stoichiometry don't seem to really add up."

The authors respond by indicating that: As we now mention in the manuscript, as opposed to other collagens, collagen VI was reported to be insensitive to metalloproteases, but sensitive to serine hydrolases, consistent with the lysosomal degradation proposed here."

This is an idea that is not supported by other work and the authors themselves have not provided evidence to support this. Data are needed to underpin this remarkable assertion.

Please accept our apologies, we have now added references to the study that reports that collagen VI is sensitive to serine hydrolases but not metalloproteases [ref 1 below], and the one that indicates that Collagen VI appears to be endocytosed and degraded by lysosomes [2]. This study also postulates that collagen VI degradation requires the activity of collagenase, cysteine and serine proteases. Also $\alpha 3$ (VI) chain, but not the other chains, has been shown to be sensitive to degradation by MMP2 [3]. Thus while full length trimeric collagen VI appear protected from MMPs digestion, this might not apply to individual chains.

The authors go on to say: "Also the analysis of the mouse uteri indicates that metalloproteases are still active, potentially even more active, explaining why other collagens do not appear to accumulate."

What is the evidence for this statement?

Our observations in figure 3cd indicate that metalloproteases (MMP2 and 14) are more active. Regarding the effect of collagen VI on metalloproteases activity, we now quote the study that shows that $\alpha 2$ (VI) binds to the latent form of several metalloproteases and thus modulates their activity [4]. Thus accumulation of collagen VI as observed in HFS patient nodules could lead to MMP sequestration and reduced collagenase activity over time.

Later in the same part of the response, the authors say:

"The activity of the metalloproteases is however likely to vary considerably with the age of the

fibrotic tissue. We only have access to "old" nodules, i.e. when they are so big that they need to be removed to improve the patients quality of life. The massive accumulation of matrix in these nodules is likely to trap various molecules and affect their structure and/or activity."

The notion that the nodules may trap other molecules and, in some way, hinder remodeling processes is again, a remarkable and potentially interesting assertion. What is the evidence for this?

The notion that nodules might trap certain molecules is at this stage a speculation based on the above mentioned ability of collagen VI to bind latent forms of metalloproteases and the fact that HFS nodule size increase with time and ECM keeps accumulating in *Antxr2* KO mice uteri. This is now mentioned in the concluding remarks.

2) In the previous review, I indicate that: "If CMG2 is indeed going to be put forward as a specific, collagen VI receptor, then more convincing and detailed experiments are needed to show receptor binding and specificity than are provided here. Connected to this idea is the need to provide structural data and binding motifs by which CMG2 could provide high affinity binding to discrete domains of collagen VI (but not other collagen types)."

No substantive data have been provided that address these points, and this is, I believe, fundamental to the mechanism which they are proposing. This should be addressed with new data.

In the revised version of the manuscript we showed additional data supporting that CMG2 is a receptor for collagen VI. Altogether we show that 1) the CMG2 vWA domain can bind collagen VI triple helical domain, 2) collagen VI triggers CMG2 phosphorylation, 3) collagen VI binding triggers β -arrestin recruitment; 4) CMG2 mediates lysosomal degradation of collagen VI.

The comment of the reviewer might contain a semantic issue: does the ability of a protein to bind a protein make it a receptor for this protein? We would argue not necessarily, for us the term receptor includes the notion of a response to binding, which we have only seen for collagen VI.

Regarding binding affinity, this is an interesting question which we believe goes beyond the scope of this manuscript. Indeed, whether the receptor is a low or a high-affinity receptors has no impact on the present conclusions.

Our prediction is however that affinity should be low, as observed for several integrin-collagen interactions. Indeed, if the affinity of collagen VI for CMG2 were high, it might leads to depletion of collagen VI from the ECM.

We wish to mention that for several months already we have attempted to perform Plasmon Resonance experiments to probe the interaction between the CMG2 vWA domain and commercially available collagen VI. Unfortunately, despite our attempts, we have not been able to attach collagen VI to the chips. While Plasmon Resonance experiments have been reported for other collagens, non exist regarding collagen Vi triple helical domains, possibly

due to similar technical issues.

References

1. Kielty CM, Lees M, Shuttleworth CA, Woolley D. Catabolism of Intact Type VI Collagen Microfibrils: Susceptibility to Degradation by Serine Proteinases. *Biochem Biophys Res Commun.* 1993 Mar;191(3):1230–6.
2. Everts V, Korper W, Niehof A, Jansen I, Beertsen W. Type VI collagen is phagocytosed by fibroblasts and digested in the lysosomal apparatus: Involvement of collagenase, serine proteinases and lysosomal enzymes. *Matrix Biol.* 1995 Oct;14(8):665–76.
3. Myint E, Brown DJ, Ljubimov AV, Kyaw M, Kenney MC. Cleavage of human corneal type VI collagen alpha 3 chain by matrix metalloproteinase-2. *Cornea.* 1996 Sep;15(5):490–6.
4. Freise C, Erben U, Muche M, Farndale R, Zeitz M, Somasundaram R, et al. The alpha 2 chain of collagen type VI sequesters latent proforms of matrix-metalloproteinases and modulates their activation and activity. *Matrix Biol.* 2009 Oct;28(8):480–9.

Reviewers' comments:

Reviewer #1 (Remarks to the Author):

the authors have addressed my major concerns. the results are now supporting the conclusions drawn.

Reviewer #2 (Remarks to the Author):

I do appreciate that the authors have performed TEM immunogold labeling to convincingly show the increase in Collagen VI deposition in the uterine extracellular matrix of the uterus of the Antxr2^{-/-} mice. However, neither Figure 2b nor Supplementary Figure 4b show controls. These controls are not mentioned in the Materials and Methods neither. The authors should have performed negative controls (with no primary antibodies) and the same experiments in wild type mice. In absence of these key controls and relative quantification of the results, it is impossible to conclude that there is a striking increase in Collagen VI deposition in the Antxr2^{-/-} mice. In addition, the number of mice used for TEM analysis should be added in the Figure Legend or in Materials and Methods.

Reviewer #3 (Remarks to the Author):

All looks good

Reply to reviewer

I do appreciate that the authors have performed TEM immunogold labeling to convincingly show the increase in Collagen VI deposition in the uterine extracellular matrix of the uterus of the Antxr2^{-/-} mice. However, neither Figure 2b nor Supplementary Figure 4b show controls. These controls are not mentioned in the Materials and Methods neither. The authors should have performed negative controls (with no primary antibodies) and the same experiments in wild type mice. In absence of these key controls and relative quantification of the results, it is impossible to conclude that there is a striking increase in Collagen VI deposition in the Antxr2^{-/-} mice. In addition, the number of mice used for TEM analysis should be added in the Figure Legend or in Materials and Methods.

The secondary antibody that we routinely use for EM is highly specific. To illustrate this we have now performed and added images in the supplementary figure showing that both in WT and KO mice, there is no unspecific labeling (we chose an image that shows at least one gold particle). The specificity of the antibodies, primary and secondary, is further supported by the fact that collagen VI labeling is exclusively found extracellularly, and in the KO mice mostly absent from collagen fibers (Fig. 2c, upper panels).

We have now split the previous figure 2 into two figures, current fig. 2 that shows morphological analyses and fig. 3 that shows western blot and mRNA analyses. In figure 2, we now first show low magnification EM images, which illustrate that the uterus of KO is almost exclusively composed of ECM, with very few cells. We then show a higher magnification that illustrates that the structure of this ECM is strongly altered and not fibrillar as the ECM in WT uteri. In panel c, we show immunogold labeling of collagen VI. The upper panels illustrate that labeling is always extracellular. To illustrate this, we chose one of the rare regions of the KO mouse uterus that actually has a cell. The lower panels in Fig. 2c show that collagen VI staining is observed in the ECM of both WT and KO mice. Interestingly in WT mice, Collagen VI is associated with collagen fibers, while in the KO mice, fibers are rare, and the collagen VI staining is highly abundant on non-fibrillar structures.

The combined observations that collagen VI is only extracellular, is abundant in the ECM and that the uteri of KO mice have far more ECM than WT mice indicate that there is altogether more collagen VI in the uteri of KO mice than that of WT mice.

We chose not to perform quantification because we believe that counting gold particles on high magnification images is not the appropriate method to quantify the abundance of a protein at the tissue level. Indeed while there might be somewhat more collagen VI per gram of volume of ECM, the overall increase in collagen VI is mostly due to the fact that there is more ECM.

The number of mice used for EM analysis, TEM and immunogold labeling, have now been added to the legend.

REVIEWERS' COMMENTS:

Reviewer #2 (Remarks to the Author):

The authors have now answered my comments in a satisfactory manner.